

# Role of CO₂, climate and land use in regulating the seasonal amplitude increase of carbon fluxes in terrestrial ecosystems: a multimodel analysis

Fang Zhao[1,2], Ning Zeng[1,3], Ito Akihiko[4], Ghassam Asrar[5], Pierre Friedlingstein[6], Atul Jain[7], Eugenia Kalnay[1], Etsushi Kato[8], Charles D. Koven[9], Ben Poulter[10], Rashid Rafique[5], Stephen Sitch[6], Shijie Shu[7], Beni Stocker[11], Nicolas Viovy[12], Andy Wiltshire[13], Sonke Zaehle[14]

[1]Department of Atmospheric and Oceanic Science, University of Maryland, College Park, MD 20742, USA
[2]Potsdam Institute for Climate Impact Research, Telegraphenberg, 14412 Potsdam, Germany
[3]Earth System Science Interdisciplinary Center, University of Maryland, College Park, MD 20742, USA
[4]Center for Global Environmental Research, National Institute for Environmental Studies, 305-0053 Tsukuba, Japan
[5]Joint Global Change Research Institute, Pacific Northwest National Laboratory, College Park, MD 20742, USA
[6]University of Exeter, Exeter EX4 4QF, UK
[7]Department of Atmospheric Sciences, University of Illinois, Urbana, IL 61801, USA
[8]Global Environment Program Research & Development Division, the Institute of Applied Energy (IAE), 105-0003 Japan
[9]Earth Sciences Division, Lawrence Berkeley National Laboratory, Berkeley, CA 94720, USA
[10]Institute on Ecosystems and Department of Ecology, Montana State University, Bozeman, MT 59717, USA
[11]Climate and Environmental Physics, Physics Institute, University of Bern, 3012 Bern, Switzerland
[12]Laboratoire des Sciences du Climat et de l'Environnement, CEA CNRS UVSQ, 91191 Gif-sur-Yvette, France
[13]Met Office Hadley Centre, Exeter EX1 3PB, United Kingdom
[14]Biogeochemical Integration Department, Max Planck Institute for Biogeochemistry, P.O. Box 10 01 64, 07701 Jena, Germany

*Correspondence to*: Fang Zhao (fzhaozhao@pik-potsdam.de)

**Abstract.** We examined the net terrestrial carbon flux to the atmosphere ($F_{TA}$) simulated by nine models from the TRENDY dynamic global vegetation model project during 1961-2012 for its seasonal cycle and amplitude trend. While some models exhibit similar phase and amplitude compared to atmospheric inversions, with spring drawdown and autumn rebound, others tend to rebound early in summer. The model ensemble mean underestimates the magnitude of the seasonal cycle by 40% compared to atmospheric inversions. Global $F_{TA}$ amplitude increase (19±8%) and its decadal variability from the model ensemble are generally consistent with constraints from surface atmosphere observations. However, models disagree on attribution of this long-term amplitude increase, with factorial experiments attributing 83±56%, −3±74% and 20±30% to rising CO₂, climate change and land use/cover change, respectively. Seven out of the nine models suggest that CO₂ fertilization is a stronger control—with the notable exception of VEGAS, which attributes approximately equally to the three factors. Generally, all models display an enhanced seasonality over the boreal region in response to high-latitude warming, but a





negative climate contribution from part of the Northern Hemisphere temperate region, and the net result is a divergence over climate change effect. Six of the nine models show land use/cover change amplifies the seasonal cycle of global $F_{TA}$: some are due to forest regrowth while others are caused by crop expansion or agricultural intensification, as revealed by their divergent spatial patterns. We also discovered a moderate cross-model correlation between $F_{TA}$ amplitude increase and increase in land carbon sink ($R^2$=0.61). Our results suggest that models can show similar results in some benchmarks with different underlying mechanisms, therefore the spatial traits of $CO_2$ fertilization, climate change, and land use/cover changes are crucial in determining the right mechanisms in seasonal carbon cycle change as well as mean sink change.

## 1 Introduction

The amplitude of $CO_2$ seasonal cycle, largely controlled by vegetation growth and decay in Northern Hemisphere (NH) (Hall et al., 1975; Pearman and Hyson, 1980; Bacastow et al., 1985; Randerson et al., 1997; Heimann et al., 1998; Graven et al., 2013), is a good indicator of terrestrial ecosystem dynamics. Since 1958, atmospheric $CO_2$ measurements at Mauna Loa, Hawaii have tracked a 15% rise in the peak-to-trough amplitude of the detrended $CO_2$ seasonal cycle (Zeng et al., 2014), suggesting an enhanced ecosystem activity due to changes in the strength of ecosystem's production, respiration and shift in the timing of their phases (Randerson et al., 1997). In addition, some evidence suggests a latitudinal gradient in $CO_2$ amplitude increase in the NH, with larger increase at Pt. Barrow, Alaska (0.6% $y^{-1}$) than at Mauna Loa (0.32% $y^{-1}$) (Randerson et al., 1999; Graven et al., 2013). Previous studies have attempted to attribute the long-term $CO_2$ amplitude increase to stimulated vegetation growth under rising $CO_2$ and increasing nitrogen deposition (Bacastow et al., 1985; Sillen and Dieleman, 2012; Reich and Hobbie, 2013). Another possible explanation offered is the effect of warmer climate, especially in boreal and temperate regions, on the lengthening of growing season, enhanced plant growth (Keeling et al., 1996; Keenan et al., 2014), vegetation phenology (Thompson, 2011), ecosystem composition and structure (Graven et al., 2013). The agricultural green revolution due to widespread irrigation, increasing management intensity and high-yield crop selection, could also contribute to the dynamics of the $CO_2$ seasonal amplitude (Zeng et al., 2014; Gray et al., 2014). Even though these studies are helpful in understanding the role of $CO_2$, climate and land use/cover changes, the detailed understanding of the relative contribution of these factors still remains unclear.

Dynamic vegetation models are useful tools not only in understanding the contribution of various mechanisms but also offering insights on how terrestrial ecosystems respond to external changes. Attribution on the role of $CO_2$, climate and land use has been attempted with a single model (Zeng et al., 2014), but comprehensive multi-model assessment efforts are still lacking. Two important questions must be addressed in such effort, namely, whether the models can simulate observed $CO_2$ amplitude increase, and to what extent their factorial attributions agree. For the first question, the CMIP5 Earth system models seem to be able to simulate the amplitude increase measured at the Mauna Loa and Point Barrow surface stations (Zhao and Zeng, 2014), however they underestimate significantly the amplitude increase compared to upper air (3-6 km) observations (Graven et al., 2013). It is possible that uncertainty in vertical mixing in atmospheric transport models (Yang et al., 2007),



instead of dynamic vegetation models themselves, causes the severe underestimation of upper air $CO_2$ amplitude increase. For the second question, in a unique modeling study conducted by McGuire et al. (2001), both $CO_2$ fertilization and land use/cover changes were found to contribute to $CO_2$ amplitude increase at Mauna Loa, but the four models disagreed on the role of climate and the relative importance of the factors they studied. Since then, no published study has explored the reliability of models'

simulation of seasonal carbon cycle and quantified the relative contribution of various factors affecting it.

An important trait of the three main factors (i.e. $CO_2$, climate and land use/cover change) we consider in this study is their different regional influence. Rising $CO_2$ would likely enhance productivity in all ecosystems. Climate warming may affect high latitude ecosystems more than tropical and subtropical vegetation, and droughts would severely affect plant growth in water-limited regions. Similarly, the effect of land use/cover change may be largely confined to agricultural fields and places

with land conversion, mostly in mid latitude regions. Because of their different spatial traits, it is possible to determine which factor is most important with strategically placed observations. Forkel et al. (2016) recently derived a latitudinal gradient of $CO_2$ amplitude increase based on $CO_2$ observational data, which would provide strong support that high latitude warming is the most important factor. However, with only two sites north of 60N, the robustness of the result is limited. In lieu of additional observational evidence, as a first step, it is necessary to investigate how the models represent the regional patterns of seasonal

cycle change of carbon flux.

A number of recent studies have addressed different aspects of the seasonal amplitude. For example, the latitudinal gradient of $CO_2$ seasonal amplitude was used as benchmark in assessing the performance of JSBACH model (Dalmonech and Zaehle, 2013; Dalmonech et al., 2015). Based on a model intercomparison project—MsTMIP (Huntzinger et al., 2013; Wei et al., 2014), Ito et al. (2015) focused on examining the relative contribution of $CO_2$, climate and land use/cover changes, but

little model evaluation was performed. In order to further explore and understand the seasonal fluctuation of carbon fluxes, a more comprehensive study including both the model evaluation and factorial analysis is needed. The TRENDY model intercomparison project provides a nice platform for such analysis (Sitch et al., 2015). Site-level model-data comparison of seasonal carbon fluxes has been performed extensively in Peng et al. (2015) for the first synthesis of TRENDY models. Using both the second synthesis of TRENDY models simulations and observations, in this study we aim to achieve two main goals:

1) Assess how well the models simulate the climatological seasonal cycle and seasonal amplitude change of the carbon flux against a number of observational based datasets ($CO_2$ observations and atmospheric inversions); 2) Analyze the relative contribution from the three main factors ($CO_2$ fertilization, climate and land use/cover change) to the seasonal amplitude increase, both at the global and regional level.

## 2 Method

### 2.1 Terrestrial Ecosystem Models and TRENDY Experiment Design

Monthly net biosphere production (NBP) simulations for 1961-2012 from nine TRENDY models participating in the Global Carbon Project (Le Quéré et al., 2014) were examined (Table 1). Three experiments were designed in the TRENDY project to





differentiate the role of $CO_2$, climate and land use (Table 2). We primarily evaluated results from the S3 experiment, where the models are driven by time-varying forcing data (Appendix A). In addition, we also used results from the S1 and S2 experiments.

## 2.2 Observations and observational based estimates

In light of the large difference in C[4]MIP models' sensitivity to $CO_2$ change (Friedlingstein et al., 2013), it is essential to evaluate if the terrestrial biosphere models are able to capture important features of $CO_2$ seasonal cycle. The scarcity of observational constraints, especially the lack of long-term continuous observational records, limits our capacity to fully evaluate the dynamic processes in terrestrial ecosystem models. Nevertheless, in this study we make a first-order approximation on the evolution of global $CO_2$ seasonal cycle, using limited $CO_2$ observation data. Following Zeng et al. (2014),

monthly Mauna Loa record from 1961 to 2012 and a global monthly $CO_2$ index for the period of 1981-2012 were retrieved from NOAA's ESRL (www.esrl.noaa.gov/gmd/ccgg/trends/). Details on the data processing, choice of stations and quality control procedures in deriving the global $CO_2$ index (globally averaged $CO_2$ concentration) can be found in Thoning et al. (1989) and Masarie and Tans (1995).

A direct comparison with fluxes from the process-based models are monthly gridded fluxes from atmospheric

inversions, which combine measured atmospheric $CO_2$ concentration at multiple sites across the globe with atmospheric transport driven by meteorological data. Two representative inversions, Jena (Jena81 and Jena99, Rodenbeck et al., 2003) and the CarbonTracker (Peters et al., 2007), are included for comparison (Appendix B). For an exhaustive intercomparison of the atmospheric inversions, please refer to Peylin et al. (2013).

## 2.3 Calculating the seasonal cycle and its amplitude change

All monthly NBP and inversion derived fluxes are first resampled (box-averaging, conserving mass) to a uniform $0.5° \times 0.5°$ global grid in unit of $kgC\ m^{-2}\ y^{-1}$. For the TRENDY model simulations, we further define net carbon flux from the land to the atmosphere ($F_{TA}$), which simply reverses the sign of NBP, so that positive $F_{TA}$ indicates net carbon release to the atmosphere, and negative indicate net carbon uptake. $F_{TA}$ represents the sum of residual land sink and land use emission, including fluxes from ecosystem production and respiration, fire, harvest, etc., although some model may not simulate all the processes.

Changes in global atmospheric $CO_2$ concentration then equal to $F_{TA}$ plus ocean-atmosphere flux and fossil fuel emission. For inversion-derived fluxes, only terrestrial ecosystem fluxes are used (bio optimized flux plus fire flux for carbon tracker), which are conceptually similar to $F_{TA}$ except that atmospheric transport is included. Atmospheric transport can significantly affect local carbon fluxes (Randerson et al., 1997), however, the impact is limited on global and large zonal band totals.

The seasonal amplitude of Mauna Loa Observatory or global $CO_2$ growth rate and fluxes from model simulations and

inversions are computed with a curve fitting package called CCGCRV from NOAA/ESRL (http://www.esrl.noaa.gov/gmd/ccgg/mbl/crvfit/crvfit.html). This package first filtered out the high-frequency signals with a series of internal steps involving polynomial and harmonic fitting, detrending and band-pass filtering, and then the amplitude





is defined as the difference between each year's maximum and minimum. For the latitudinal plots only, we simply use maximum and minimum of each year as seasonal amplitude without first filtering the data. Previous studies (Randerson et al., 1997; Graven et al., 2013) have established that $F_{TA}$ accounts for most of seasonal amplitude change from atmospheric $CO_2$, and Mauna Loa $CO_2$ record is considered to represent the evolution of global mean $CO_2$ well (Kaminski et al., 1996). Therefore,

similar to our earlier work (Zeng et al., 2014), we evaluated the amplitude change of modeled $F_{TA}$ with Mauna Loa $CO_2$, ESRL's global $CO_2$ and the atmospheric inversions, to assess whether the models are able to capture both the global trend and latitudinal patterns. For relative amplitude changes, we compute the multi-model ensemble mean after deriving the time series (relative to their 1961-1970 mean) from individual model simulations, so that models with large amplitude change would not have a huge effect on the ensemble mean. Additionally, global and regional mean seasonal cycles over 2001-2010 between

the models and inversions are compared. We further compared the seasonal amplitude of zonally averaged $F_{TA}$ from TRENDY and atmospheric inversions. To smooth out minor variations but ensure similar phase in aggregation, we first resampled $F_{TA}$ into 2.5 °resolution, then summed over latitude bands for the 2001-2010 mean $F_{TA}$ seasonal cycle.

### 2.4 Factorial analyses

Relative amplitude for 1961-2012 (relative to 1961-1970 mean seasonal amplitude) from the experiments S1, S2 and S3,

respectively, are calculated using the CCGCRV package for each model, and a linear trend (in % y$^{-1}$) is determined for that period. We assume that models simulate these three main effects fairly linear, which is likely plausible as supported by previous sensitivity experiment results (Zeng et al., 2014). We use relative amplitude for percentage change to minimize impacts of some differing implementation choices like climate data in S1 ($CO_2$) among the models. Therefore, the S2 ($CO_2$+Climate) results show a trend that is the sum of $CO_2$ and climate effects, and the S3 ($CO_2$+Climate+Land Use/Cover) simulations include

trends from time-varying $CO_2$, climate and land use/cover change (abbreviated as LandUse for text and figures). With this linear assumption, effect of $CO_2$, climate and land use/cover are then quantified as the trend for S1, trend of S2 minus S1 trend, and trend of S3 minus S2 trend, respectively.

### 2.5 Spatial attribution

Spatial attribution of global $F_{TA}$ amplitude change can be difficult due to the phase difference at various latitudes. For example,

the two amplitude peaks at Northern and Southern subtropics caused by monsoon movements are largely out of phase, and the net contribution to global $F_{TA}$ amplitude increase after their cancelation is small (Zeng et al., 2014). To quantify latitudinal and spatial contributions for each model, a unique quantity—$F_{k\_A}^{i}$, difference between the maximum month ($i\_max$) and the minimum month ($i\_min$) of model $i$'s global $F_{TA}$, based on model $i$'s 2001-2010 mean seasonal cycle was defined in Eq. (1):

$$F_{k\_A}^{i} = F_{k\_A(i\_max)}^{i} - F_{k\_A(i\_min)}^{i} ,   \hspace{2cm} (1)$$





The subscript $k$ denotes index for each latitudinal band or spatial grid, and $A$ is index of year, ranging from 1961 to 2012. $F_{k\_A}^{i}$ could be quite different for each model: for VEGAS, $F_{k\_A}^{i}$ is $F_{TA}$ in November ($i\_max$ = 11) minus $F_{TA}$ in July ($i\_min$ = 7) in year $A$, and for LPJ, $F_{k\_A}^{i}$ is $F_{TA}$ in March ($i\_max$ = 3) minus $F_{TA}$ in June ($i\_min$ = 6) in year $A$. The indexes $i\_max$ and $i\_min$ are fixed for each model, as summarized in Table 3. For all three experiments, $F_{k\_A}^{i}$ is computed each year in 1961-2012 and

at every latitude band or spatial grid ($k$), and then the trends of $F_{k\_A}^{i}$ are calculated. The spatial aggregation of the resulted latitudinal-depended trends would then approximately equal to trend of global $F_{TA}$ maximum-minus-minimum seasonal amplitude.

## 3 Results

### 3.1 Mean seasonal cycle of $F_{TA}$

Four of the nine models (CLM4.5BGC, LPX-Bern, ORCHIDEE and VEGAS) simulate a mean global $F_{TA}$ seasonal cycle of similar amplitude and phase compared with the Jena99 and CarbonTracker inversions (Figure 1, Table 3). The other five models have much smaller seasonal amplitude than inversions, and the shape of the seasonal cycle is also notably different. As a result, models' ensemble global $F_{TA}$ has seasonal amplitude of 26.1 PgC y$^{-1}$ during 2001-2010, about 40% smaller than the inversions (Figure 4 inset, Table 3). The model ensemble annual mean $F_{TA}$ (residual land sink plus land use emission) is

−1.1 PgC y$^{-1}$ for 2001-2010, 30% smaller than the inversions (Table 3). In some models (ISAM, JULES, and LPJ for the Northern Temperate region in Figure 2) $F_{TA}$ rebounds back quickly, resulting in a late summer $F_{TA}$ maximum. The mid-summer rebound is unlikely a model response to pronounced seasonal drought after 2000, as it is persistent in the mean seasonal cycle over every decade since 1961. A probable cause is the strong exponential response of soil respiration to temperature increase, which may lead to heterotopic respiration higher than NPP in summer. For example, the HadCM3LC that employs

TRIFFID, an earlier version of JULES3.2 used in this study, is found to have a large mid-summer peak carbon release over temperate North America (Cadule et al., 2012). Alexandrov (2014) shows that both the amplitude underestimation and phase shift of $F_{TA}$ seasonal cycle can be improved by increasing water use efficiency, decreasing Q10 value, and increasing the share of quickly decaying litterfall. Another probable factor is the simulation of plant phenology. With the help of remote sensing data, better phenology in model simulation has been shown to improve seasonal cycle simulation of carbon flux (Forkel et al.,

2014). Additionally, the effect of carbon release from crop harvest is considered. If harvested carbon is the main cause for the mid-summer rebound in some models, the rebound should be much less pronounced for the S2 (constant 1860 land use/cover) experiment, given that cropland area in 1860 is less than half of the 2000 level. However, based on the comparison between the S2 and S3 experiments over global and northern temperate (major crop belts) $F_{TA}$ seasonal cycle (Figure S1 and S2), the impact of harvested carbon flux is unlikely to explain the mid-summer rebound. This is probably due to modeling efforts to

prevent the sudden release of harvested carbon. Instead, carbon release of harvested products and/or their residuals is usually




either spread over 12 months (i.e., LPJ, LPX-Bern, OCN, ORCHIDEE) or enters soil litter carbon pool (i.e., ISAM) for subsequent decomposition over time.

   TRENDY models and inversions agree best over the boreal region (Figure 2a). While underestimating the global seasonal cycle, LPJ and VISIT both simulate similar boreal $F_{TA}$ amplitude as inversions. In addition to ORCHIDEE and
VEGAS, LPJ and LPX-Bern also simulate maximum $CO_2$ drawdown in July for the boreal region, same as the inversions, while the other five models have the $F_{TA}$ minimum in June. Large model spread is present for the Northern temperate region especially in summer. Both inversions and models agree marginally over the phase of the $F_{TA}$ seasonal cycle in the tropics. The Northern and Southern tropics show seasonal cycles that are largely out of phase except for LPJ (Figure 2c, d), due to the seasonal movement of tropical rain belt in the Inter-Tropical Convergence Zone (ITCZ). The Southern extra-tropics exhibit
even smaller $F_{TA}$ amplitude due to its small biomass, and most models and inversions indicate a maximum $F_{TA}$ around July, opposite in phase to its NH counterpart.

   The latitudinal pattern of the multi-model median $F_{TA}$ amplitude is remarkably similar to the inversions (Figure 3). A notable feature is the large seasonality over NH mid-high latitude region driven by temperature contrast between winter and summer. The model median also captures the two subtropical maxima around 10N and 15S that are caused by tropical monsoon
movement. The main difference between the TRENDY models and the two inversions is in the tropics and SH, where several models (JULES, LPJ, OCN and especially ORCHIDEE) show much higher amplitude than the inversions. Seasonal amplitude over 37-45N and 53-60N is also larger from TRENDY models than the inversions. A majority of the models display larger amplitude in the tropics and Northern temperate regions. Only three models (ISAM, JULES and OCN) exhibit underestimation of seasonal amplitude in the north of 45N. Because of phase difference among the models and at different latitudinal bands,
for spatial and cross-model aggregated carbon fluxes, the seasonal amplitude is reduced. Similarly, analyses by Peng et al. (2015) with an earlier set of TRENDY models (Sitch et al., 2015) show approximately equal number of models overestimating and underestimating carbon flux compared to flux sites north of 35N. However, once the carbon fluxes of different phases are transported and mixed, seven out of nine models underestimate the $CO_2$ seasonal amplitude compared to $CO_2$ site measurements (Peng et al., 2015). Note that even at the same latitude band, factors like monsoons, droughts, and spring snow
melt, etc. could lead to longitudinal difference in the phase of seasonal cycle (Figure S3 and S4).

## 3.2 Temporal evolution of $F_{TA}$ seasonal amplitude

The seasonal amplitude of global total $F_{TA}$ from the TRENDY model ensemble for 1961-2012 shows a long-term rise of 19±8%, with large decadal variability (Figure 4). Similarly, the seasonal amplitude of $CO_2$ at Mauna Loa increases by 15±3% (0.85±0.18 ppm) for the same period. This amplitude increase appears mostly as an earlier and deeper drawdown during the
spring and summer growing season, mostly in June and July (Table 3, Figure 4 inset). Changes in trend of yearly minima (indicating peak carbon uptake) and yearly maxima (dominated by respiration) contribute 91±10% and 9±10% to the $F_{TA}$ amplitude increase, respectively. Gurney and Eckels (2011) suggest trend in respiration increase is more important, but they averaged all months instead of maxima and minima in their amplitude definition. The multi-model ensemble mean tracks some



characteristics of the decadal variability reflected by the Mauna Loa record: stable in the 1960s, rise in the 1970-1980s, rapid rise in the early 2000s, and decrease in most recent 10 years. Strictly speaking, Mauna Loa $CO_2$ data are not directly comparable with simulated global $F_{TA}$, because this single station is also influenced by atmospheric circulation, as well as fossil fuel emissions and ocean–atmosphere fluxes. Nevertheless, the comparison on long-term amplitude trend is still valuable because

the Mauna Loa Observatory data constitute the only long-term record, and it is generally considered representative of global mean $CO_2$ (Heimann, 1986; Kaminski et al., 1996). The global total $CO_2$ index ($CO_{2GLOBAL}$) and $F_{TA}$ from three atmospheric inversions are also included in the comparison. All data (Jena81, $CO_{2MLO}$, $CO_{2GLOBAL}$) show a decrease in seasonal amplitude in the late 1990s, possibly related to drought in the Northern Hemisphere mid-latitude regions (Zeng et al., 2005b; Buermann et al., 2007), and about half of the models also exhibit similar trend (Figure 7). Details on models' $F_{TA}$ global and regional

changes in 2001-2010 compared to 1961-1970 are listed in Table 4.

### 3.3 Attribution of global and regional $F_{TA}$ seasonal amplitude

Models agree on increase of global $F_{TA}$ seasonal amplitude during 1961-2012 (Figure 5). By computing the ratios between amplitude trends from rising $CO_2$, climate change and land use/cover change with the total trend for each model, we find the effect of varying $CO_2$, climate and land use/cover contribute 83±56%, −3±74% and 20±30% to the simulated global $F_{TA}$

amplitude increase. All models simulate increasing amplitude for total $F_{TA}$ in the boreal (50-90N) and Northern temperate (23.5-50N) regions, and most models also indicate amplitude increase in the Northern (0-23.5N) and Southern tropics (0-23.5S) (Figure 6). There is a less agreement on the sign of amplitude change among the models in the Southern extra-tropics (23.5-90S). Individual model's global and regional trends of $F_{TA}$ amplitude attributable to the three factors ($CO_2$, climate and land use/cover) are listed in Table S1. For most models, latitudinal contribution to global $F_{TA}$ amplitude (computed with $F_{k\_A}^i$)

shows that the pronounced mid-high latitude maxima in the NH dominate the simulated amplitude increase over 1961-2012 (Figure 8, red dashed line for S3 results). All models also indicate a negative contribution from at least part of the Northern temperate region.

### 3.3.1 The rising CO2 factor

Seven of the nine models suggest that $CO_2$ fertilization effect is most responsible for the increase in the amplitude of global

$F_{TA}$, while VEGAS attribute it approximately equal among the three factors (Figure 5). The $CO_2$ fertilization effect alone seems to cause most of the amplitude increase in a majority of models, with notable contribution from climate change and land use/cover change in CLM4.5BGC and VEGAS (Figure 7). The effect of rising $CO_2$ appears to be slightly negative for JULES, possibly reflecting uncertainty associated with experiment design (randomized climate is used to drive JULES). For each model, rising $CO_2$ in the boreal, Northern temperate and the Southern extra-tropics leads to a similar trend (Figure 6). The

magnitude of this trend may indicate each model's differing strength for $CO_2$ fertilization. This is possibly due to similar phases of $F_{TA}$ seasonal cycle within the three regions that are mainly driven by climatological temperature contrast. The positive amplitude trend in the carbon flux of the Northern and Southern tropics from $CO_2$ fertilization is similar, and they



likely would cancel out each other because their seasonal cycles are largely out of phase. Latitudinal contribution analyses reveal that trends in the Northern mid-high latitude is the main contributor to global $F_{TA}$ amplitude increase when considering $CO_2$ fertilization effect alone (Figure 8, blue line).

### 3.3.2 The climate change factor

The effect of climate change on $F_{TA}$ amplitude is mixed: five models (OCN, LPJ, LPX-Bern, ORCHIDEE and ISAM) suggest climate change acts to decrease the $F_{TA}$ amplitude, and four models (JULES, VISIT, CLM4.5BGC and VEGAS) suggest it is an increasing effect (Figure 5). The high-latitude greening effect is evident in six out of nine models (Figure 6), contributing on average 29% of boreal amplitude increase. The latitudinal contribution analyses (Figure 8) also suggest that warming induced high latitude "greening" effect is present in all models, but this positive contribution only exhibits a wide range of

influence in about half of the models (CLM4.5BGC, JULES, VEGAS and VISIT). The latitudinal patterns also reveal that, once climate change is considered, the contribution from the Northern temperate region around 40N shifts to negative in all models. In the Northern temperate (23.5-50N) region, climate change alone would decrease the $F_{TA}$ amplitude except for JULES and LPJ (Figure 6), possibly related to mid-latitude drought (Buermann et al., 2007). This is consistent with findings by Schneising et al. (2014), who observed a negative relationship between temperature and seasonal amplitude of $xCO_2$ from

both satellite measurements and CarbonTracker during 2003-2011 for the Northern temperate zone. The negative contribution from the temperate zone counteracts the positive boreal contribution, suggesting the net impact from climate change on $F_{TA}$ amplitude may not be as significant as previously suggested. With changing climate introduced, some models exhibit similar characteristics of decadal variability in global $F_{TA}$ amplitude (Figure 7). OCN and ORCHIDEE appear to be especially sensitive to the climate variations after the 1990s, resulting in a decrease in $F_{TA}$ amplitude. It is also apparent from the time series figure

that the strong increasing trend of $F_{TA}$ amplitude from climate change in JULES is mostly due to the sharp rise from early 1990s to early 2000s, suggesting some possible model artifact (Figure 7). The effect of climate change is more mixed in both tropics and the Southern extra-tropics.

### 3.3.3 The land use/cover change factor

Six of the nine models show that land use/cover change leads to increasing global $F_{TA}$ amplitude (Figure 5). Land use/cover

change appears to amplify $F_{TA}$ seasonal cycle in boreal and Northern temperate regions for most models. For some models (VEGAS, CLM4.5BGC and OCN), this effect is especially pronounced in the Northern temperate region where most of the global crop production takes place (Figure 6). Note that the effect of land use/cover change includes two parts: one is the change of land use practice without changing the land cover type; the other is the change of land cover, including crop abandonment etc. VEGAS simulates time-varying management intensity and crop harvest index, which is an example of

significant contribution from land use change (Zeng et al., 2014). For many other models, crop is treated as generic managed grasslands (i.e., CLM4.5BGC, LPJ), and land cover change is possibly the more important factor. During 1961-2012, large cropland areas were abandoned in the Eastern U.S. and central Europe, and forest regrowth often followed. New cropland





expanded in the tropics and South America, Midwest U.S., East and central North Asia and the Middle East. How such change affect the global $F_{TA}$ amplitude is determined by the productivity and seasonal phase of the old and new vegetation covers. For CLM4.5BGC, JULES, LPJ and ORCHIDEE, enhanced vegetation activity from growing forest in these regions contribute positively to global $F_{TA}$ amplitude increase (Figure 9). In contrast, for LPX-Bern, VISIT, and VEGAS in the Eastern U.S., loss

of cropland leads to decrease in the amplitude. Additional cropland in the Midwest U.S. and East and central North Asia contribute negatively to $F_{TA}$ amplitude trend for JULES, LPJ and ORCHIDEE. These regions however, are major zones contributing the amplification of global $F_{TA}$ for LPX-Bern, OCN, VEGAS and VISIT. One mechanism mentioned previously is agricultural intensification in VEGAS: in fact, $CO_2$ flux measurements over corn fields in the U.S. Midwest show much larger seasonal amplitude than over nearby natural vegetation (Miles et al., 2012). Similarly, although croplands are treated as

generic grassland, they still receive time-varying and spatially explicit fertilizer input in OCN (Zaehle et al., 2011). Another plausible mechanism is irrigation, which can alleviate adverse climate impact from droughts, and crops may have a stronger seasonal cycle than the natural vegetation they replace in these regions. The overall effect of land use/cover change for each model therefore, is often the aggregated result over many regions that can only be revealed by spatially explicit patterns. When examining the latitudinal contribution only (Figure 8), CLM4.5BGC, LPX-Bern, OCN and VEGAS are quite similar, even

though the spatial patterns reveal CLM4.5BGC is very different from the other three models (Figure 9). For JULES, LPJ and ORCHIDEE a significant part of land use/cover change contribution comes from the tropical zone (Figure 8).

## 4 Discussion and conclusion

Our results show a robust increase of global and regional (especially over the boreal and Northern temperate regions) $F_{TA}$ amplitude simulated by all TRENDY models. During 1961-2012, TRENDY models' ensemble mean global $F_{TA}$ relative

amplitude increase (19±8%). Similarly, the $CO_2$ amplitude also increases (15±3%) at Mauna Loa for 1961-2012. This amplitude increase mostly reflects the earlier and deeper drawdown of $CO_2$ in the NH growing season. The models in general, especially the multi-model median, simulate latitudinal patterns of $F_{TA}$ mean amplitude that is similar with the atmospheric inversions results. Their latitudinal patterns capture the temperature driven seasonality from the NH mid-high latitude region and the two monsoon driven subtropical maxima, although the magnitude or extent vary. Despite the general agreements

between the models' ensemble amplitude increases and the limited observation-based estimates, considerable model spread are noticeable. Five of the nine models considerably underestimate the global mean $F_{TA}$ seasonal cycle compared to atmospheric inversions, and peak carbon uptake takes place one or two months too early in seven of the nine models. The seasonal amplitude of model ensemble global mean $F_{TA}$ is 40% smaller than the amplitude of the atmosphere inversions. In contrast to the divergence in simulated seasonal carbon cycle, atmospheric inversions in Northern temperate and boreal regions

are well constrained: 11 different inversions agree on July $F_{TA}$ minimum in the Northern Hemisphere (25-90N), with no more than 20% difference in amplitude (Peylin et al., 2013).



The simulated amplitude increase is found to be due to a larger $F_{TA}$ minimum associated with a stronger ecosystem growth. Over the historical period, global mean carbon sink is also increasing over time, suggesting a possible relationship between seasonal amplitude and the mean sink (Randerson et al., 1997; Zhao and Zeng, 2014; Ito et al., 2015). The increasing trend of $CO_2$ amplitude, dominated by increasing trend of $F_{TA}$ amplitude, has been interpreted as evidence for steadily

increasing net land carbon sink (Keeling et al., 1995; Prentice et al., 2000). However, the increasing amplitude could also arise from (climatically induced) increased phase separation of photosynthesis and respiration, e.g., due to warming-induced earlier "greening" (Myneni et al., 1997). For the nine models, we found a moderate relationship between enhanced mean land carbon sink and the seasonal amplitude increase similar to reported results by in Zhao and Zeng (2014), with an R-squared value of 0.61 (Figure 10). There might be some possibility in constraining change in land carbon sink with changes in observed $CO_2$

seasonal amplitude, however extra caution should be given when interpreting this global-scale cross-model correlation, as there could be important regional differences that cancel out in aggregated global values. Further research is needed to explore the mechanisms behind such relationship at continental-scale, where more data from well calibrated $CO_2$ monitoring sites, and data on air-sea fluxes and atmospheric vertical transport could better constrain carbon balance (Prentice et al., 2001). Changes of residual land carbon sink estimates are also shown (Figure 10), with the caveat that it is not directly comparable with

simulated net carbon sink increase, if there is a trend in simulated carbon flux changes associated with land cover conversion (deforestation, crop abandonment, etc.). Additionally, the decadal changes of residual and net land carbon sink are far from linear, instead a sudden increase in mean land uptake occurred in 1988 (Sarmiento et al., 2010; Beaulieu et al., 2012; Rafique et al., 2016). With the aid of atmospheric transport, $CO_2$ amplitude trends at remote sites have benchmarking potential to constrain the models, especially with more observations and improved understanding of vegetation dynamics at regional level

in the near future.

Models with a strong mean carbon sink (for example JULES and OCN) can have relatively weak seasonal amplitude, and the LPX-Bern model shows no carbon sink despite having a strong $F_{TA}$ seasonality. Based on data from Table 8 of the Global Carbon Budget report (Le Quéré et al., 2014), the net land carbon sink for 2000-2009 is estimated to be 1.5±0.7 PgC $y^{-1}$ (assuming Gaussian errors). Four models (JULES, OCN, VEGAS and VISIT) examined in this study are within the

uncertainty range of this budget-based analysis. In spite of their similar mean land carbon sink, the shape of their $F_{TA}$ seasonal cycle differs. While VEGAS also shows a similar seasonal carbon cycle compared to inversions, the other three models exhibit an unrealistically long carbon uptake period with half the amplitude as the inversions. July and August are the only two months with net carbon release for JULES, whereas OCN and VISIT both have a long major carbon uptake period from May to September. Given that the mean global and regional $F_{TA}$ seasonal cycles are relatively well constrained in the northern extra-

Tropics, and they can serve as benchmark for terrestrial models (Heimann et al., 1998; Prentice et al., 2001). Insights gained from analyzing modeled seasonal amplitude of carbon flux may help to understand the considerable model spread found in the mean global carbon sink for the historical period (Le Quéré et al., 2015), which is possibly due to varied model sensitivity to different mechanisms (Arora et al., 2013). Examining details of models' mechanical difference could also help to better assess the different future projections on both the magnitude and direction of global carbon flux (Friedlingstein et al., 2006, 2013).



Unlike many previous studies that focused on comparing season cycle at individual $CO_2$ monitoring stations (Randerson et al., 1997; Peng et al., 2015), we studied the global and large latitudinal bands, Such quantities often demonstrate well-constrained seasonality that is relatively robust against uncertainty from atmospheric transport, fossil fuel emission, biomass burning etc.. We found greater uncertainty for the tropics and Southern extra-tropics regions where atmospheric $CO_2$ observations are relatively sparse. Tropical ecosystems are also heavily affected by biomass burning, however some models used in this study do not include fire dynamics. For models that simulate fire ignition/suppression, they are also varied by structure and complexity of fire-related processes, and many of them are prognostic (Poulter et al., 2015). It is not clear how fire would affect the $F_{TA}$ seasonal cycle at global scale, and recent sensitivity study shows only minor differences among fire and "no fire" scenarios in $CO_2$ seasonal cycle at several observation stations (Poulter et al., 2015). These uncertainties however, are unlikely to affect our main conclusions because of limited contribution of tropics to global $F_{TA}$ amplitude increase. Another possibly important factor is the impact from increased nitrogen deposition, which may have been include in the "$CO_2$ fertilization" effect for some models with full nitrogen cycle (Table 1), however this can only be explored in future studies, as the TRENDY experiment design does not separated out the nitrogen contribution.

Our factorial analyses highlight fundamentally differential control from rising $CO_2$, climate change and land use/cover change among the models, with seven out of nine models indicating major contribution (83±56%) to global $F_{TA}$ amplitude increase from the $CO_2$ fertilization effect. The strength of $CO_2$ fertilization varies among models, but for each model its magnitude in the boreal, Northern temperate and Southern extra-tropics regions is similar. Models are split regarding the role of climate change, as compared with the models ensemble mean (−3±74%). Regional analyses show that climate change amplifies the boreal $F_{TA}$ seasonal cycle but weakens the seasonal cycle for other regions according to most models. By examining latitudinal trends from $F_{k\_A}^i$, we found all models indicate a negative climate contribution over the mid-latitudes, where droughts might have reduced ecosystem productivity. This negative effect offsets the high latitude "greening" over high latitudes, which in some models result in a net negative climate change impact on global $F_{TA}$ amplitude. Such mechanism cast doubt on whether climate change is the main driver of the global $F_{TA}$ amplitude increase. Land use/cover change, according to majority of the models, appears to amplify the global $F_{TA}$ seasonal cycle (20±30%), however the mechanisms seem to differ for different models. Conversion to/from cropland could either increase or decrease the seasonal amplitude, depending on how models simulate the seasonal cycle of cropland compared to natural vegetation it replaces/precedes. For the same pattern of increasing amplitude, the underlying causes could include irrigation that mitigates negative climate effect, agricultural management practices and other mechanisms.

Overall, this study is largely helpful to enhance our understanding on role of $CO_2$, climate change and land use/cover change in regulating the seasonal amplitude of carbon fluxes. Especially, models disagreement in spatial pattern of carbon flux amplitude help to identify optimal locations for additional CO2 observations in the north. However, this work can be further improved through utilizing the $CO_2$ seasonal cycle and its amplitude at different locations as indicators to diagnose model behaviors. To achieve this, it is necessary to apply atmosphere transport on the simulated net carbon flux, along with ocean and fossil fuel fluxes, which would allow direct comparison with observed $CO_2$ amplitude change. In doing so, it is possible





that model may overestimate $CO_2$ amplitude increase at most $CO_2$ observation stations, if the simulated $CO_2$ fertilization effect is too strong.

## Appendices

### A. Environmental drivers for TRENDY

For observed rising atmospheric $CO_2$ concentration, the models use a single global annual (1860-2012) time series from ice core (before 1958: Joos and Spahni, 2008) and the National Oceanic and Atmospheric Administration (NOAA)'s Earth System Research Laboratory (after 1958: monthly average from Mauna Loa and South Pole $CO_2$, south pole data is constructed from the 1976-2014 average if not available). For climate forcing, the models employ 1901-2012 global climate data from the Climate Research Unit (CRU, version TS3.21, http://www.cru.uea.ac.uk; or CRU-National Centers for Environmental Prediction (NCEP) dataset, version 4 from N. Viovy 2011, unpublished data, available online at http://dods.ipsl. jussieu.fr/igcmg/IGCM/BC/OOL/OL/CRU-NCEP/)) at monthly (or interpolate to finer temporal resolution for individual models) temporal resolution and $0.5\,^{\circ} \times 0.5\,^{\circ}$ spatial resolution. For land use/cover change history data, the models adopt either gridded yearly cropland and pasture fractional cover from the History Database of the Global Environment (HYDE) version 3.1 (http://themasites.pbl.nl/tridion/en/themasites/hyde/, Klein Goldewijk et al., 2011), or the dataset including land use history transitions from L. Chini based on the HYDE data.

### B. Atmospheric Inversions

The Jena inversion is from the Max Planck Institute of Biogeochemistry, v3.7 at $5\,^{\circ} \times 5\,^{\circ}$ spatial resolution (http://www.bgc-jena.mpg.de/christian.roedenbeck/download-CO2/, Rodenbeck et al., 2003), including two datasets abbreviated as Jena81 for the period of 1981–2010 using CO2 data from 15 stations, and Jena99 using 61 stations for 1999–2010. Another inversion-based dataset used is the CarbonTracker, version CT2013B from NOAA/ESRL at $1\,^{\circ} \times 1\,^{\circ}$ spatial resolution (http://www.esrl.noaa.gov/gmd/ccgg/carbontracker/, Peters et al., 2007) for the period of 2000–2010, which integrates flask samples from 81 stations, 13 continuous measurement stations and 9 flux towers, and the surface fluxes from land and ocean carbon models as prior fluxes. These two inversion-based datasets are vastly different in their approach in inversion algorithm, choice of atmospheric data, transport model and prior information (Peylin et al., 2013). For example, to minimize the spurious variability introduced by changes in availability of observations, the Jena inversion provides multiple versions with different record length, each only use records covering its full period (for example, Jena99 includes more stations than Jena81, but with a shorter period). The CarbonTracker however, opt for assimilating all quality-controlled data (with outliers removed) favoring a higher spatial resolution in estimated carbon fluxes. Therefore, we chose these two inversions to capture to some extent the uncertainty in atmospheric inversions.

**Author contribution**





F. Zhao and N. Zeng designed the study and F. Zhao carried it out. S. Sitch and P. Friedlingstein designed and coordinated TRENDY experiments. TRENDY modelers conducted the simulations. F. Zhao wrote the paper with input from all authors.

**Acknowledgement**

This study was funded by NOAA, NASA and NSF. Partial financial support for this study was also provided by a Pacific Northwest National Laboratory Directed Research and Development project. We thank the TRENDY coordinators and participating modeling teams, NOAA ESRL, and Jena/CarbonTracker inversion teams. TRENDY model results used in this study may be obtained from S. Sitch (email: s.a.sitch@exeter.ac.uk).

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

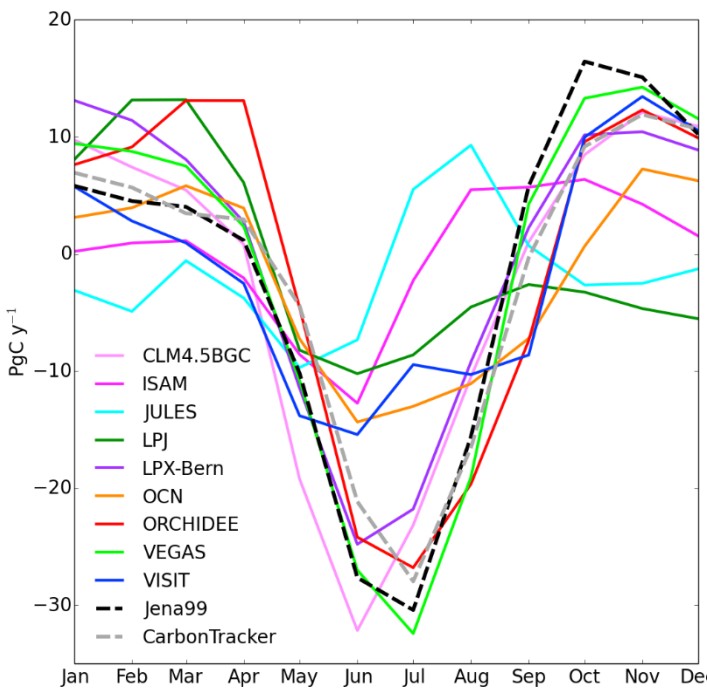

**Figure 1. Mean seasonal cycle of global net carbon flux from nine TRENDY models (S3 experiment) and two inversions, Jena99 and**
**CarbonTracker, averaged over 2001-2010.**





**Figure 2. Mean seasonal cycle of net carbon flux totals over boreal (50-90N), Northern temperate (23.5-50N), Northern tropics (0-23.5N), Southern tropics (0-23.5S) and Southern extra-tropics (23.5-90S) from nine TRENDY models and two inversions, Jena99 and CarbonTracker, averaged over 2001-2010.**





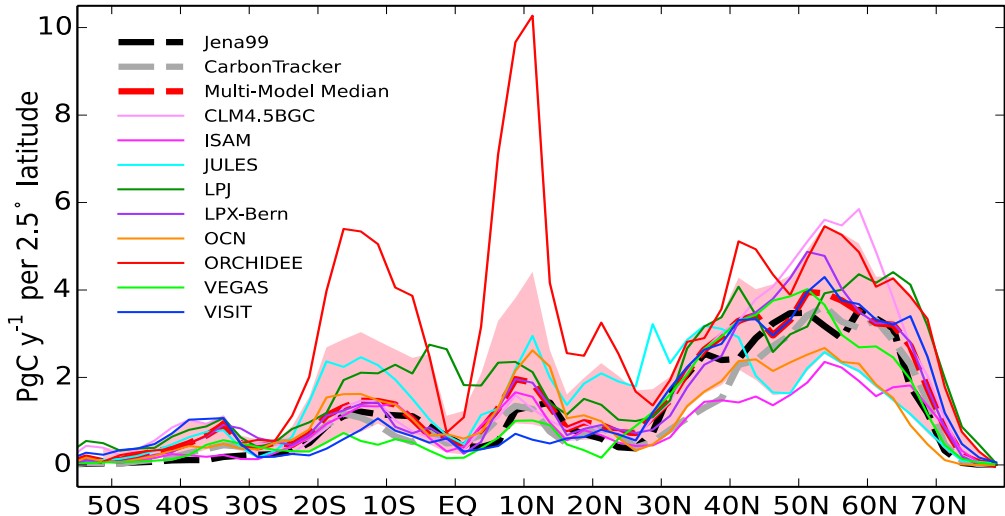

**Figure 3. Latitudinal dependence of the seasonal amplitude of land-atmosphere carbon flux from the TRENDY multi-model median (red line, and the pink shade indicates 10 to 90 percentile range of model spread), two atmospheric $CO_2$ inversions, Jena99 (black dashed) and CarbonTracker (grey dashed), and each individual model (thin line). Fluxes are first resampled to $2.5° \times 2.5°$, then summed over each $2.5°$ latitude bands (PgC $y^{-1}$ per $2.5°$ latitude) for the TRENDY ensemble and inversions.**



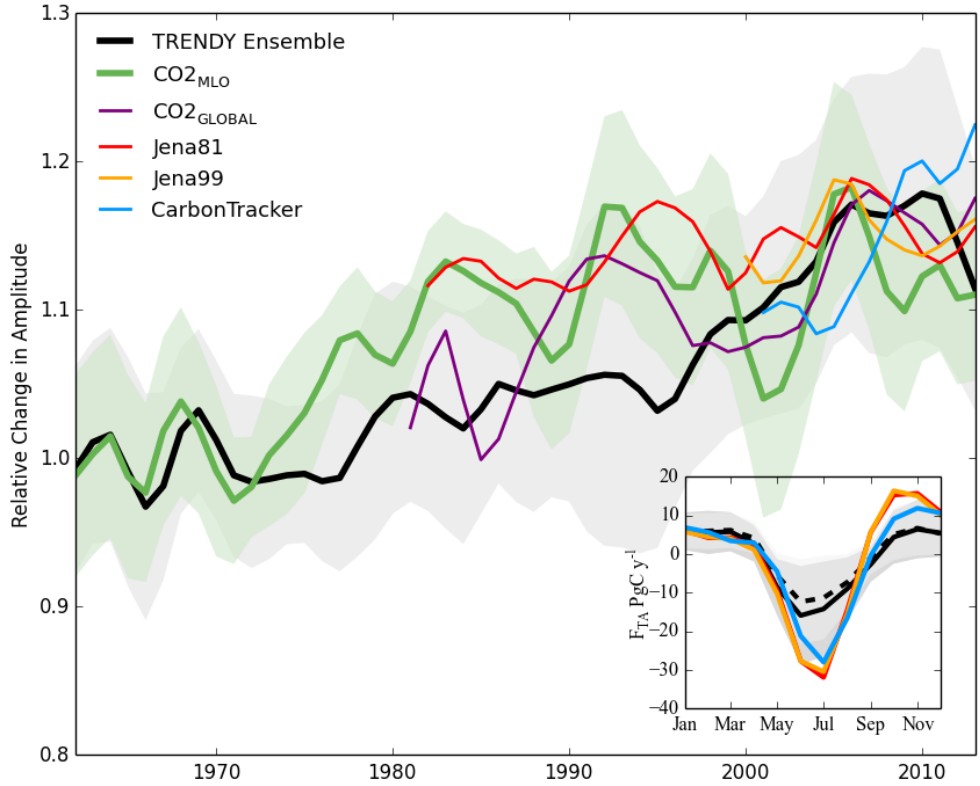

**Figure 4. Trends for seasonal amplitude of TRENDY simulated multi-model ensemble mean land-atmosphere carbon flux $F_{TA}$ (black), of MLO $CO_2$ mixing ratio ($CO2_{MLO}$, green) and global $CO_2$ mixing ratio ($CO2_{GLOBAL}$, purple), and of $F_{TA}$ from atmospheric inversions of Jena81 (red), Jena99 (orange), and CarbonTracker (blue). The trends are relative to the 1961-70 mean for the TRENDY**
5  **ensemble and Mauna Loa $CO_2$, and the other time series are offset to have the same mean as the TRENDY ensemble for the last ten years (2003-2012). A 9-year Gaussian smoothing (Harris, 1978) removes inter-annual variability for all time series, and its 1-σ standard deviation is shown for $CO2_{MLO}$ (green shading). Note that the grey shading here instead indicates 1-σ models' spread, which is generally larger than the standard deviation of TRENDY ensemble's decadal variability. Inset: average seasonal cycles of models' ensemble mean $F_{TA}$ (PgC $y^{-1}$) for the two periods: 1961-1970 (dashed, lighter grey shade indicates 1-σ model spread) and**
10  **2001-2010 (solid, darker grey shade indicates 1-σ model spread), revealing enhanced $CO_2$ uptake during spring/summer growing season. Mean seasonal cycles global $F_{TA}$ from the atmospheric inversions for 2001-2010 are also shown (same color as the main figure) for comparison.**



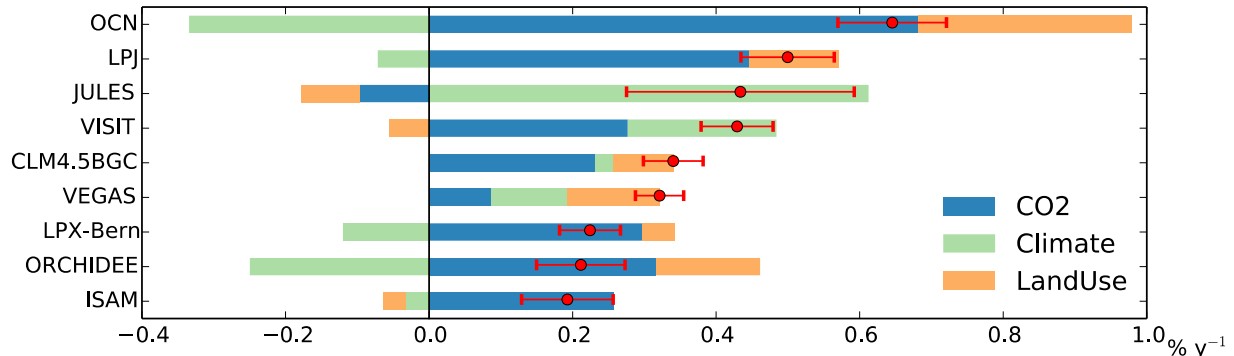

**Figure 5. Attribution of the seasonal amplitude trend of global net land carbon flux for the period 1961-2012 to three key factors of $CO_2$, climate and land use/cover. The red dots represent models' global amplitude increase of $F_{TA}$ from the S3 experiment, and error bars indicate 1-σ standard deviation. The increasing seasonal amplitude of $F_{TA}$ is decomposed into the influence of time varying atmospheric $CO_2$ (blue), climate (light green), and land use/cover change (gold).**






**Figure 6. Attribution of the seasonal amplitude trend of regional (boreal (50-90N), Northern temperate (23.5-50N), Northern tropics (0-23.5N), Southern tropics (0-23.5S) and Southern extra-tropics (23.5-90S)) net land carbon flux for the period 1961-2012 to three key factors $CO_2$, climate and land use/cover. The red dots represent models' global amplitude increase of $F_{TA}$ from the S3 experiment. The increasing seasonal amplitude of $F_{TA}$ is decomposed into the influence of time varying atmospheric $CO_2$ (blue), climate (light green), and land use/cover change (gold).**



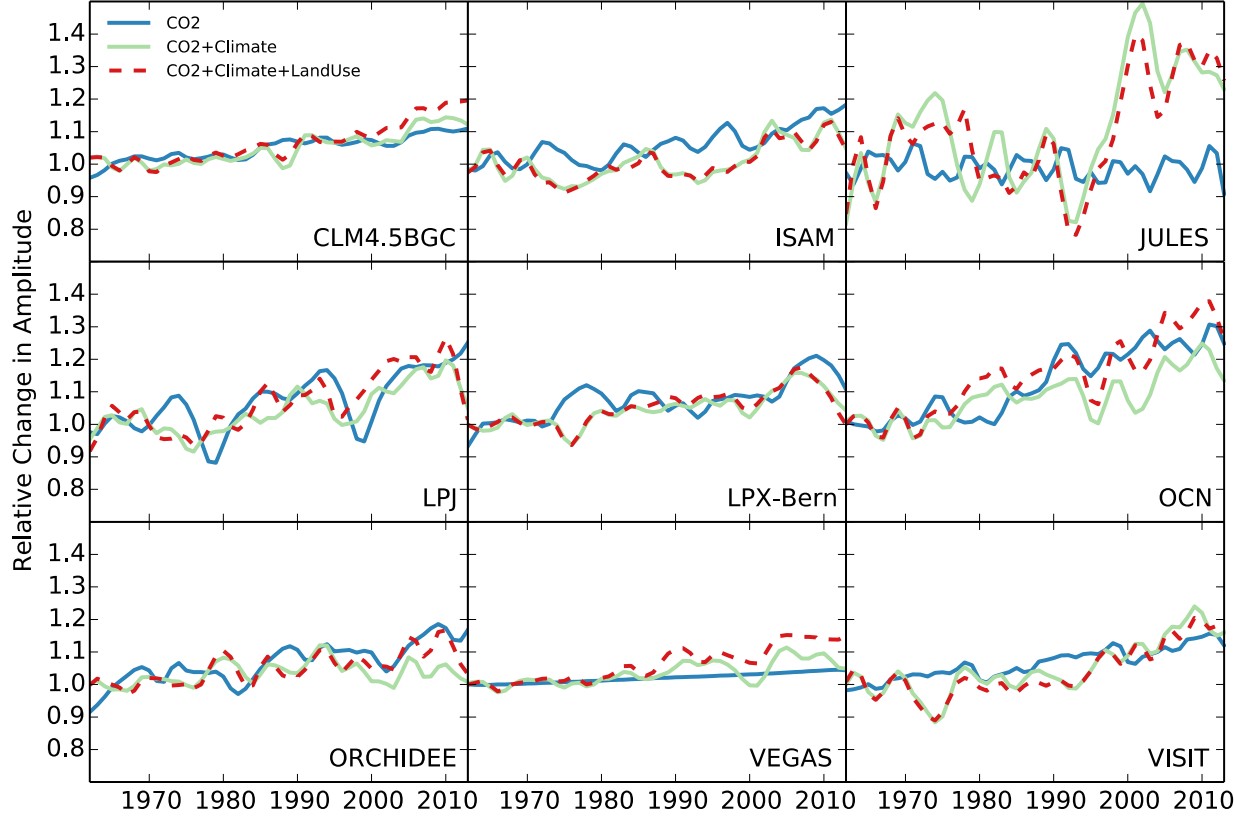

**Figure 7. Trends for seasonal amplitude of global total net carbon fluxes from S1 ($CO_2$), S2 ($CO_2$+Climate) and S3 ($CO_2$+Climate+LandUse) for each individual TRENDY model. All amplitude time series are relative to their own 1961-1970 mean amplitude.**



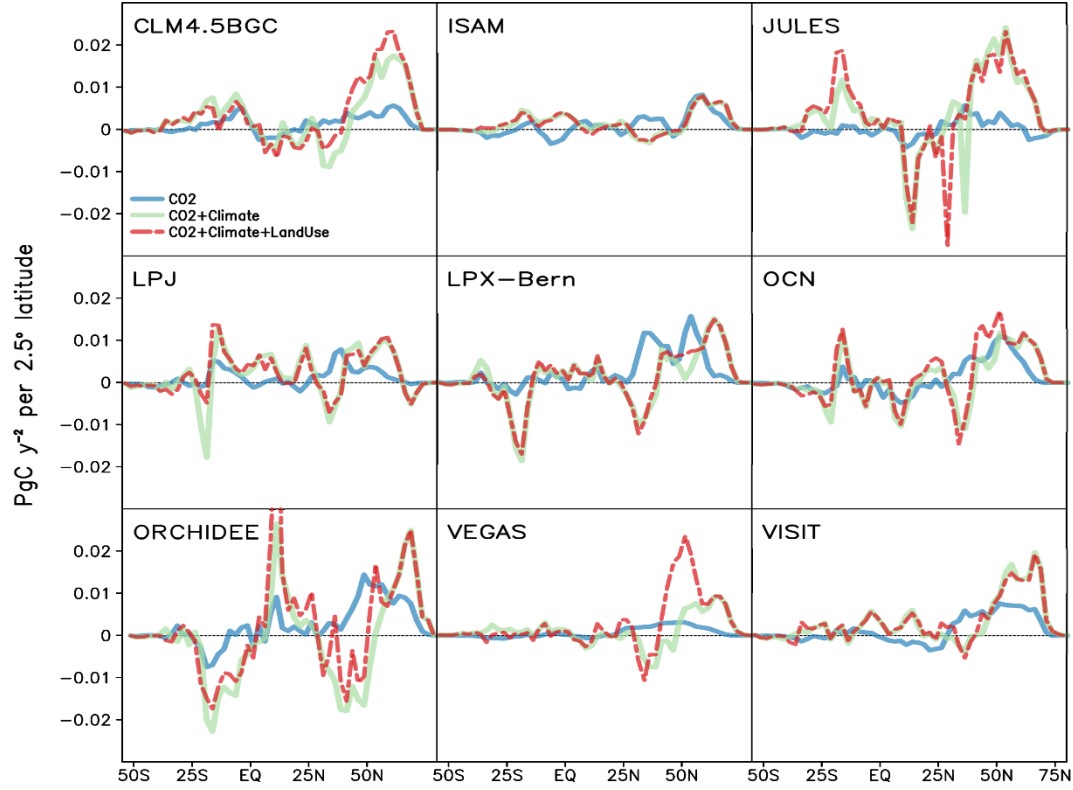

**Figure 8. Latitudinal contribution of trends for seasonal amplitude of global land-atmosphere carbon flux from TRENDY models in the three sensitivity experiments. Fluxes are summed over each 2.5 °latitude bands (PgC y$^{-1}$ per 2.5 °latitude) before computing the $F^l_{k\_A}$ (refer to Methodology section for definition). For each 2.5 °latitude band, trend is calculated for the period 1961-2012.**





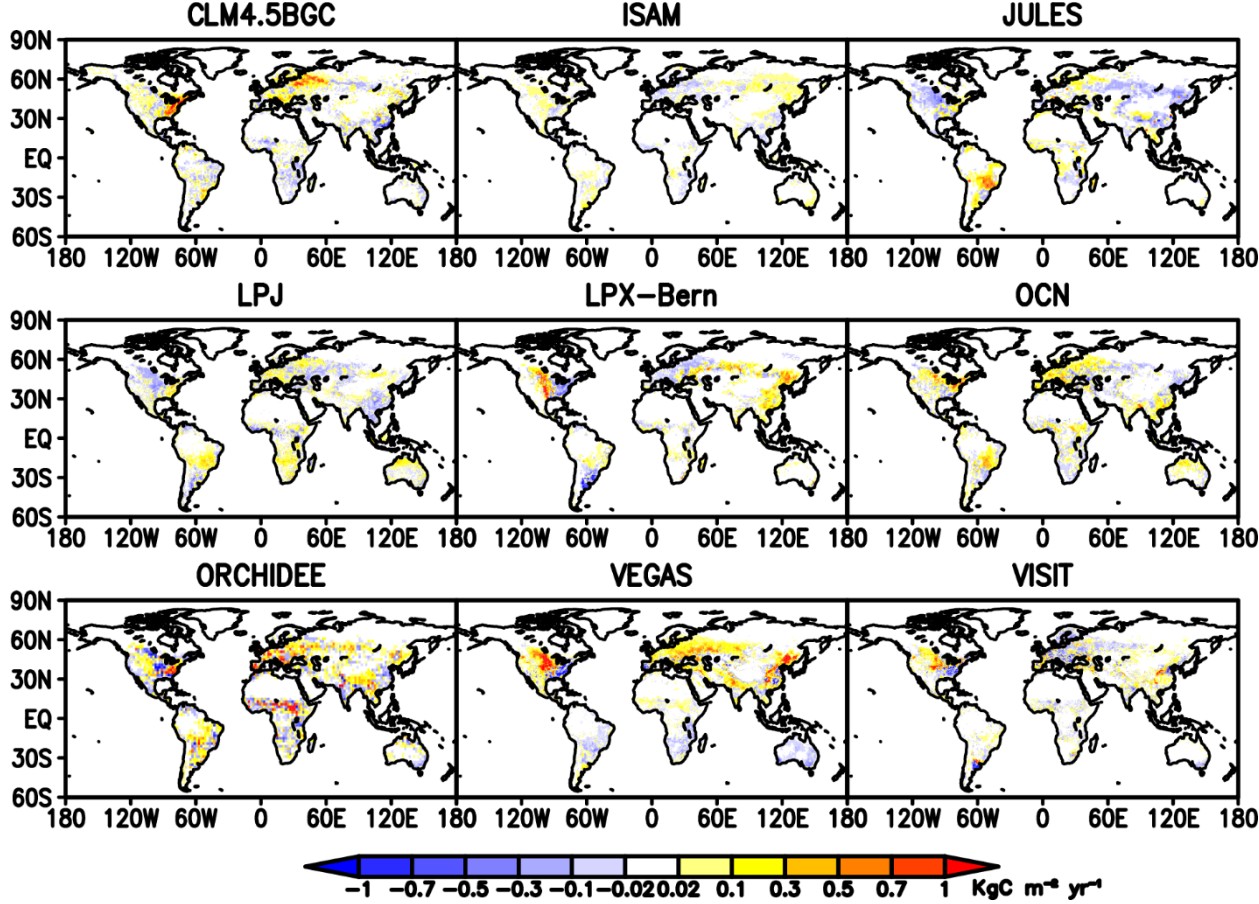

**Figure 9. Contribution from land use/cover change on trends in the seasonal amplitude of global land-atmosphere carbon flux. For each spatial grid, the trend is computed as trends of the $F^i_{k\_A}$ (refer to Methodology section for definition) in the S2 experiment (changing $CO_2$ and climate) subtracted by trends in S1 (changing $CO_2$ only).**





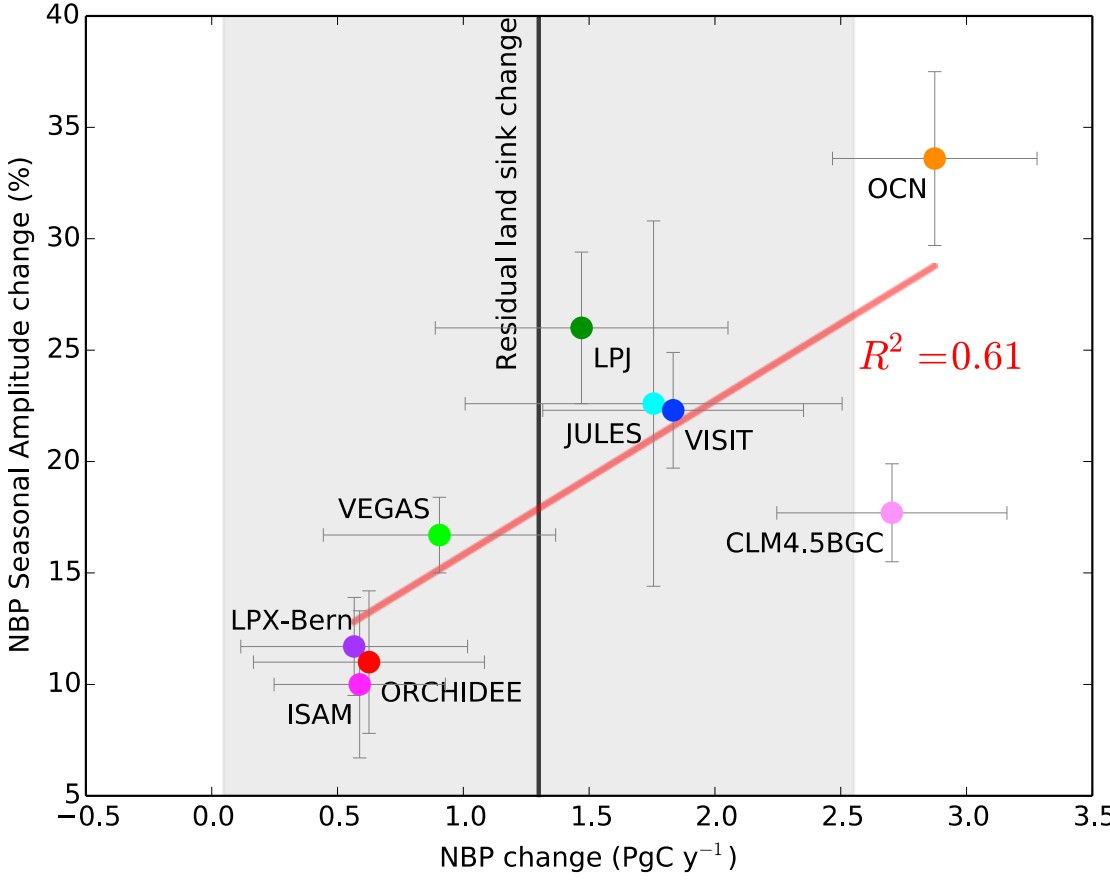

**Figure 10. Relationship between the increase in net biosphere production (NBP, equal to −F_TA) and increase in NBP seasonal amplitude (as in Figure 4's red dots), for 1961-2012 period for nine TRENDY models. Error bars indicate the standard errors of the trend estimates. Increase in residual land sink is estimated by taking the difference between two residual land sinks, over 2004-2013 and 1960-1969 (an interval of 44 years), as reported in Le Quéré et al. (2015). This difference is then scaled by 52/44 (to make it comparable with models' NBP change for this figure), which is displayed in black vertical line and shade (error add in quadrature, assuming Gaussian error for the two decadal residual land sinks, then also scaled). The cross-model correlation (R²=0.61, p < 0.05) suggests that a model with a larger net carbon sink increase is likely to simulate a higher increase in NBP seasonal amplitude.**



**Table 1. Basic information for the nine TRENDY models used in this study.**

| Model Name | Abbreviation | Spatial Resolution | Nitrogen Cycle | Fire Simulation | Harvest Flux | Reference |
|---|---|---|---|---|---|---|
| Community Land Model 4.5 | CLM4.5BGC | 1.25 °×0.94 ° | yes | yes | no | Oleson et al. (2013) |
| ISAM | ISAM | 0.5 °×0.5 ° | yes | no | yes | Jain et al. (2013) |
| Joint UK Land Environment Simulator | JULES | 1.875 °×1.25 ° | no | no | no | Clark et al. (2011) |
| Lund-Potsdam-Jena | LPJ | 0.5 °×0.5 ° | no | yes | yes | Sitch et al. (2003) |
| LPX-Bern | LPX-Bern | 0.5 °×0.5 ° | yes | yes | yes | Stocker et al. (2014) |
| O-CN | OCN | 0.5 °×0.5 ° | yes | no | yes | Zaehle and Friend (2010) |
| ORCHIDEE | ORCHIDEE | 2 °×2 ° | no | no | yes | Krinner et al. (2005) |
| VEGAS | VEGAS | 0.5 °×0.5 ° | no | yes | yes | Zeng et al. (2005) |
| VISIT | VISIT | 0.5 °×0.5 ° | no | yes | yes | Kato et al. (2013) |

 

**Table 2. Experimental design of TRENDY simulations.**

| Name | Time Period | Atmospheric CO2 | Climate Forcing | Land-use History** |
|------|-------------|-----------------|-----------------|--------------------|
| S1 | 1901-2012 | Time-varying | Constant* | Constant (1860) |
| S2 | | | Time-varying | |
| S3 | | | | Time-Varying |

*Constant climate state achieved by repeated or randomized or fixed climate cycles depending on each model. **Only the crop, pasture and wood harvest information are included, so "land use" in this study refers specifically to the related agricultural and forestry processes.



**Table 3. Global mean net land carbon flux, seasonal amplitude, the maximum and minimum months of $F_{TA}$ for the nine TRENDY models and their ensemble mean during 1961-1970 and 2001-2010 periods. For the later period, characteristics of the atmosphere inversions Jena99 and CarbonTracker are also listed.**

| Name | Net Carbon Flux (PgC y$^{-1}$) | | Seasonal Amplitude (PgC y$^{-1}$) | | $F_{TA}$ Minimum | $F_{TA}$ Maximum |
|---|---|---|---|---|---|---|
| | 1961-1970 | 2001-2010 | 1961-1970 | 2001-2010 | 2001-2010 | 2001-2010 |
| CLM4.5BGC | 0.1 | −2.4 | 38.4 | 44.3 | Jun | Nov |
| ISAM | 0.7 | 0.0 | 17.6 | 19.1 | Jun | Oct |
| JULES | −0.2 | −1.7 | 15.1 | 19.0 | May | Aug |
| LPJ | 1.3 | −0.6 | 18.6 | 23.4 | Jun | Mar |
| LPX-Bern | 0.6 | 0.0 | 33.0 | 37.9 | Jun | Jan |
| OCN | 0.9 | −1.8 | 16.1 | 21.6 | Jun | Nov |
| ORCHIDEE | 0.1 | −0.7 | 35.7 | 39.9 | Jul | Mar |
| VEGAS | −0.4 | −1.5 | 40.7 | 46.7 | Jul | Nov |
| VISIT | 0.2 | −1.4 | 25.3 | 28.9 | Jun | Nov |
| Ensemble | 0.4 | −1.1 | 22.4 | 26.1 | Jun | Nov |
| Jena99 | | −1.7 | | 46.8 | Jul | Oct |
| CarbonTracker | | −1.6 | | 39.9 | Jul | Nov |



Table 4. The seasonal amplitude (maximum minus minimum, in PgC y$^{-1}$) of mean net carbon flux for 2001-2010 relative to the 1961-1970 period, according to the nine TRENDY models (values are listed as percentage change in brackets, for both regions and the entire globe). The four large latitudinal regions are the same as in Figure 3: boreal (50-90N), temperate (23.5-50N), Northern tropics (0-23.5N), Southern tropics (0-23.5S), and Southern extra-tropics (23.5-90S). Values from the two inversions Jena99 and CarbonTracker are also listed for comparison.

| Name | Global | Boreal | Northern Temperate | Northern Tropics | Southern Tropics | Southern extra-Tropics |
|---|---|---|---|---|---|---|
| CLM4.5BGC | 44.3(15%) | 31.9(17%) | 19.2(15%) | 7.2 (22%) | 6.5 (−2%) | 4.9(4%) |
| ISAM | 19.1 (9%) | 12.1(11%) | 7.4(13%) | 6.0(1%) | 6.9 (−8%) | 0.4(4%) |
| JULES | 19.0(26%) | 12.2(24%) | 14.3(9%) | 11.6(0%) | 11.3(11%) | 2.2(−24%) |
| LPJ | 23.4(26%) | 23.0(18%) | 14.7(11%) | 10.5(9%) | 11.8(16%) | 2.0(−12%) |
| LPX-Bern | 37.9(15%) | 26.9(10%) | 19.3(6%) | 8.3(9%) | 4.6 (−6%) | 4.2(15%) |
| OCN | 21.6(34%) | 12.3(33%) | 11.1(23%) | 9.7 (17%) | 8.3(3%) | 2.0(14%) |
| ORCHIDEE | 39.9(12%) | 23.4(14%) | 19.1(5%) | 22.7(9%) | 18.7(2%) | 1.4(37%) |
| VEGAS | 46.7(15%) | 22.3(17%) | 24.7(10%) | 4.0 (11%) | 3.4 (12%) | 2.1(6%) |
| VISIT | 28.9(14%) | 22.9(12%) | 15.6(8%) | 3.4(9%) | 3.2(1%) | 3.1(18%) |
| Ensemble | 26.1(17%) | 18.0(19%) | 12.4(15%) | 8.0(8%) | 4.9(−3%) | 2.1(13%) |
| | | | | | | |
| Jena99 | 46.8 | 23.3 | 21 | 8.2 | 8.5 | 1.5 |
| CarbonTracker | 39.9 | 26.5 | 16.3 | 5.3 | 5.8 | 2.4 |