# Peer review of "Role of CO2, climate and land use in regulating the seasonal amplitude increase of carbon fluxes in terrestrial ecosystems: a multimodel analysis"

_Biogeosciences, 2016_

## Referee Comment (RC1) · Anonymous Referee #1 · 27 Apr 2016

The paper addresses relevant scientific questions within the scope of BG and presents a novel analysis of the simulations of net terrestrial carbon flux to the atmosphere produced by nine models from the TRENDY dynamic global vegetation model project. Some substantial conclusions are reached. From my point of view, the most important is that some of well-respected models underestimate the magnitude of the flux seasonal cycle compared to atmospheric inversions. This result of the study is essential for stimulating further research which are necessary for better understanding of the factors that regulate the net terrestrial carbon flux to the atmosphere.

[Figure]

**BGD**

The scientific methods and assumptions are valid and clearly outlined. However, I am not sure that the results could be easily reproducible. The data set of the outputs of TRENDY dynamic global vegetation model project, which are available at http://dgvm.ceh.ac.uk/node/9, includes simulations made with Hyland, JULES, LPJ, LPJ-GUESS, NCAR-CLM4, ORCHIDEE, OCN, SDVGM, VEGAS, whereas the study analyses the simulations made with CLM4.5BGC, ISAM, JULES, LPJ, LPX-Bern, OCN, ORCHIDEE , VEGAS, VISIT. Perhaps, it would better to clarify that the study analyses the new set of TRENDY models in the first lines of the article, and provide some information about the expected date of the data set publication at the TRENDY website.

Besides, it would be better to update some references. For example,

Alexandrov, G. a.: Explaining the seasonal cycle of the globally averaged CO2 with a carbon cycle model, Earth Syst. Dyn., 10 5(1), 63–81, doi:10.5194/esdd-5-63-2014, 2014.

could be changed to

Alexandrov, G. A.: Explaining the seasonal cycle of the globally averaged CO2 with a carbon cycle model, Earth Syst. Dyn., 5, 345–354, doi:10.5194/esd-5-345-2014, 2014

and

Sitch, S., Friedlingstein, P., Gruber, N., Jones, S. D., Murray-Tortarolo, G., Ahlström, A., Doney, S. C., Graven, H., Heinze, 20 C., Huntingford, C., Levis, S., Levy, P. E., Lomas, M., Poulter, B., Viovy, N., Zaehle, S., Zeng, N., Arneth, A., Bonan, G., Bopp, L., Canadell, J. G., Chevallier, F., Ciais, P., Ellis, R., Gloor, M., Peylin, P., Piao, S., Le Quéré, C., Smith, B., Zhu, Z. and Myneni, R.: Trends and drivers of regional sources and sinks of carbon dioxide over the past two decades, Biogeosciences Discuss., 10(12), 20113–20177, doi:10.5194/bgd-10-20113-2013, 2015.

could be changed to

Sitch, S., Friedlingstein, P., Gruber, N., Jones, S. D., Murray-Tortarolo, G., Ahlström,

[Figure]

A., Doney, S. C., Graven, H., Heinze, 20 C., Huntingford, C., Levis, S., Levy, P. E., Lomas, M., Poulter, B., Viovy, N., Zaehle, S., Zeng, N., Arneth, A., Bonan, G., Bopp, L., Canadell, J. G., Chevallier, F., Ciais, P., Ellis, R., Gloor, M., Peylin, P., Piao, S., Le Quéré, C., Smith, B., Zhu, Z. and Myneni, R.: Trends and drivers of regional sources and sinks of carbon dioxide over the past two decades, Biogeosciences, 12, 653–679, doi:10.5194/bg-12-653-2015, 2015.

I would also recommend to check carefully the list of the authors. It seems to me that "Ito Akihiko" should be changed to "Akihiko Ito" at the Page 1, line 5.

---

## Referee Comment (RC2) · Anonymous Referee #2 · 4 Aug 2016

Review of the article "Role of CO2, climate and land use in regulating the seasonal amplitude increase of carbon fluxes in terrestrial ecosystems: a multimodel analysis" by Zhao et al.

This article presents interesting results in the scope of BG. It can be published after the authors haven taken care of the minor issues stated below. Therefore, I've asked for minor revisions only. However, I'm convinced that the paper could be improved substantially by a small effort, if the discussion would be extended in the two directions I try to describe in the following:

[Figure]

The study has two parts. In the first part it is evaluated, if nine global vegetation models (from the TRENDY project) can reproduce the seasonal cycle of land carbon fluxes as derived from atmospheric inversions (figure 1 to 3). In the second part of the article these models are used to investigate trends in the seasonal cycle amplitude of land carbon fluxes. This is done by separating the contributions to the trends for three major forcing factors: rising atmospheric CO2 concentration, trends in climate, and land use/cover change (figure 5 to 9).

In view of the large uncertainties in land carbon cycle model results, I appreciate this study very much. Combining model evaluation and factorial analysis will probably provide a better understanding of the seasonal cycle carbon fluxes (as also mentioned by the authors in the introduction, page 3 line 20-21). Unfortunately, this potential is not utilized. The findings about the performance of each model are (almost) not mentioned in the discussion of the second part. For example, the results shown in figure 5 could be discussed in view of the evaluation presented in figure 1. Some questions like the following could be posed and answered: are the models that successfully simulate the seasonal cycle of carbon uptake more similar in the CO2 fertilization factor than the models that fail to reproduce the seasonal cycle? Or is this true for the climate factor? Analogously, figure 6 could be compared with figure 2.

Finally, we want to understand by such studies, why the magnitude of the seasonal cycle in atmospheric CO2 increased. Additionally, we want to confirm that vegetation models respond reasonably to global warming and increasing atmospheric CO2. The self-evident way to reach both aims is to evaluate models and then to analyse their results by taking into account this evaluation. The later step is incomplete in the current version of the text as the evaluation is not considered in the discussion of the forced simulation results.

Another important comment I would like to add concerns figure 10. This shows a moderate correlation between the change in net land carbon uptake and the increase in the amplitude of the seasonal carbon fluxes as simulated by the different vegetation models. The authors mention that this cross-model correlation may be used to constrain land carbon uptake (page 11 line 9). I think, this is the key motivation to investigate the seasonal amplitude of carbon fluxes and it should be also mentioned in the introduction. Furthermore, the authors claim for more research on observed CO2 fluxes and atmospheric transport on a regional scale to substantiate this finding (page 11, line 11-13). I agree, but the obvious next step is to further investigate the results of the model ensemble, how the correlation is simulated. A factorial analysis of the long-term carbon uptake should be performed and then compared with figure 5. Thereby, it could be specified, which factor contributes to what extent to the correlation. Probably this is beyond the scope of the article, but at least this opportunity should be mentioned. And, this kind of analysis is commonly denoted by the keyword "emergent constraint". It would be good to cite a reference, perhaps the paper of Cox et al 2013.

Minor scientific issues

- Why is the exceptional result of VEGAS concerning the CO2 factor mentioned in the abstract (page 1 line 32)? It is not a key finding.

- "is a good indicator of terrestrial ecosystem dynamics" (page 2 line 11). This is a too general statement. It does not help in the line of argument. I would skip it.

- It would be nice to mention in section 2.1 that the TRENDY model simulations are offline simulations driven by climate data (and other input like atm. CO2 concentration) and that the models are not coupled to general circulation models. Of course this can be deduced from appendix A, but it should be also clearly stated in the main text as it is important for its perceivability (e.g. the differences in the results between the models can not be due to weather noise).

- The authors assume that the models simulate the effect of CO2, climate, and land use "linearly" (page 5 line 16-22). I think, "linear" is missleading in this context. It is more about synergy terms. For example, the trend in S2 minus S1 includes the climate effect and the synergy of CO2+climate. Furthermore, I'm not convinced that the synergy

terms are all unimportant. Therefore, I propose to simply state that the climate effect and the synergy of CO2+climate together are called "climate" for simplicity in the rest of the manuscript. Without a discussion whether the synergy terms are negligible or not. And, of course, analogously for S3 minus S2.

- Please replace "Q10 value" (page 6 line 22) by "temperature dependence of heterotrophic respiration". Not everyone is familiar with this shortcut.

- Concerning the temporal trend in the seasonal amplitude in the late 90s (page 8 line 9): I can not deduce from figure 7 that half of the models exhibit a decrease, but it is obvious that the model ensemble shows an increase.

- The models agree in the general seasonal amplitude increase, but they disagree in the contribution of the climate factor as well as the land use factor to the seasonal amplitude trend (see figure 5). I think, this disagreement even in sign should be mentioned in section 3.3. In the following subsections this important fact may be overseen due to all the details stated there.

Typos etc.

page 1 line 25: replace "during 1961-2012 for its seasonal cycle and amplitude trend" by "for its seasonal cycle and amplitude trend during 1961-2012"

page 1 line 31: replace "is a stronger" by "is the strongest"

page 2 line 9: replace "of CO2 seasonal cycle" by "of atmospheric CO2 seasonal cycle"

page 2 line 25: replace "in understanding the contribution of various mechanisms" by "to disentangle effects of various mechanism"

page 3 line 1: replace "instead of dynamic vegetation models" by "instead of biases in dynamic vegetation models"

page 4 line 14: replace "A direct comparison with fluxes from process-based models are monthly" by "Fluxes from process-based models can be directly compared with

monthly"

page 4 line 29: replace "The seasonal amplitude of Mauna Loa Observatory or global CO2 growth rate and fluxes from model simulations and inversions are computed" by "The seasonal amplitude at Mauna Loa Observatory, global CO2 growth rate, and fluxes from model simulations and inversions are processed"

page 5 line 2: replace "as seasonal amplitude" by "to define the seasonal amplitude"

page 5 line 28: replace "was defined in Eq. (1):" by "is defined as:"

page 11 line 33: replace "models' mechanical difference" by "the different parametrisations of important processes in models"

page 12 line 21: replace "high latitude "greening" over high latitudes" by "high latitude "greening""

page 12 line 24: replace "to differ for different models" by "to differ between models"

page 27: replace "in the S2 experiment (changing CO2 and climate and land use/cover) substracted by trends in S1 (changing CO2 only)" by "in the S3 experiment (changing CO2, climate, and land use/cover) substracted by trends in S2 (changing CO2 and climate)"

Please proof articles. They are missing quite often.

---

## Author Response (AR1)

**Summary of Major Changes**

There are two main issues raised by the reviewers: reviewer 1 asks if results of this study are reproducible. We believe so and the entire TRENDY dataset will be available soon, but the exact website and form of publishing have not be decided. After consulting with my co-authors regarding the data availability, we have added the following in Appendix A:

Results of TRENDY models analysed in this study will be made available based on request by the end of 2016 (please contact S. Sitch at S.A.Sitch@exeter.ac.uk for further updates and details).

Reviewer 2 suggests that the evaluation in the earlier part of this paper should be considered in the discussion of the factorial analyses results. Following this suggestion, we found some interesting consistency in the four models that simulate more realistic global seasonal cycle of carbon flux. We have now incorporated these findings in the text, which connects the evaluation with the discussion of the factorial analyses.

Additionally, since the initial submission, Dr. Wiltshire (one of co-authors) from the JULES team has solved the JULES issues, and the new version of JULES simulates a much more realistic seasonal cycle. More accurate description of the original problem has been updated in the manuscript.

In the revised manuscript, we have made some changes to methods, results, discussions and conclusions sections, and appendix A, based on reviewers comments. We have also done another round of proofing and made many minor textual changes throughout the text. Specifically, the following items have been changed.

**1. Author list.** The error in one of the co-author names has been corrected. As a result, the order of co-authors (alphabetic from third author) and institution numbers have changed accordingly.

**2. Method.** The fact that the simulations analyzed are offline driven by climate and other forcings is stated from the beginning. For the factorial analysis (section 2.4), the linear assumption for the factors is replaced by stating that "climate" and "land use/cover" effects also include some synergy terms, even though we have reasons to believe these terms are likely small in many of the current generation dynamic global vegetation models.

**3. Results.**

Sect. 3.1. Updated a more accurate explanation (from Dr. Wiltshire) of results from JULES. Replaced the term "Q10" with an explanation that is easier to understand.

Sect. 3.2. Spelled out the four models that showed some decrease in the late 90s.

Sect. 3.3. Stated in the beginning that the models disagree even in sign in their contribution of the different factors. Also added one paragraph discussing the similarity of simulated amplitude increase among the models that simulate a more realistic seasonal carbon flux, relating figure 5 to figure 1.

Sect. 3.3.1. Updated a more accurate explanation (from Dr. Wiltshire) of results from JULES.

Sect. 3.3.2. Discussed the similarity in climate factor for Northern temperate region among the models that simulate a more realistic seasonal carbon flux, relating figure 6 to figure 2.

Sect. 3.3.3. Added a discussion on LPJ results, relating figure 6 to figure 2.

**4. Discussions and conclusions.** We identify the opportunity for further research, specifically a factorial analysis on long-term sink as a next step to understand which factor contributes to what extent to the correlation in Figure 10.

**5. Appendix A.** Added the way to obtain the TRENDYv2 data and the related details.

**6. Figure 9.** Corrected typo in description.

Specific point-by-point responses to each reviewer are given below.

Responses are in bold.

Reviewer #1

The paper addresses relevant scientific questions within the scope of BG and presents a novel analysis of the simulations of net terrestrial carbon flux to the atmosphere produced by nine models from the TRENDY dynamic global vegetation model project. Some substantial conclusions are reached. From my point of view, the most important is that some of well-respected models underestimate the magnitude of the flux seasonal cycle compared to atmospheric inversions. This result of the study is essential for stimulating further research which are necessary for better understanding of the factors that regulate the net terrestrial carbon flux to the atmosphere. The scientific methods and assumptions are valid and clearly outlined.

**Thank you for the positive statements.**

However, I am not sure that the results could be easily reproducible. The data set of the outputs of TRENDY dynamic global vegetation model project, which are available at http://dgvm.ceh.ac.uk/node/9, includes simulations made with Hyland, JULES, LPJ,LPJ-GUESS, NCAR-CLM4, ORCHIDEE, OCN, SDVGM, VEGAS, whereas the study analyses the simulations made with CLM4.5BGC, ISAM, JULES, LPJ, LPX-Bern, OCN,

ORCHIDEE , VEGAS, VISIT. Perhaps, it would better to clarify that the study analyses the new set of TRENDY models in the first lines of the article, and provide some information about the expected date of the data set publication at the TRENDY website.

**Thanks for raising this issue. The TRENDY project has been an important component contributing to the highly influential annual reports of Global Carbon Budget. Right now only TRENDYv1 data is publicly available online, and the data analyzed here is from TRENDYv2, which is the latest TRENDY version with S1-S3 experiment set up (the latest version with S2 and S3 only is v4). We have mentioned this on P3 in L24-26:**
**"Site-level model-data comparison of seasonal carbon fluxes has been performed extensively in Peng et al. (2015) for the first synthesis of TRENDY models. Using both the second synthesis of TRENDY models simulations and observations,"**
**The TRENDYv2 dataset will be made available on request by the end of this year. There are also plans to make this publicly available via the global carbon atlas (possibly other websites, to be arranged). We now added this sentence at the end of appendix A:**
**" Results of TRENDY models analysed in this study will be made available on request by the end of 2016 (please contact S. Sitch at S.A.Sitch@exeter.ac.uk for further updates and details)."**

Besides, it would be better to update some references. For example,
Alexandrov, G. a.: Explaining the seasonal cycle of the globally averaged CO2 with a carbon cycle model, Earth Syst. Dyn., 10 5(1), 63–81, doi:10.5194/esdd-5-63-2014, 2014.
could be changed to
Alexandrov, G. A.: Explaining the seasonal cycle of the globally averaged CO2 with a carbon cycle model, Earth Syst. Dyn., 5, 345–354, doi:10.5194/esd-5-345-2014, 2014
and
Sitch, S., Friedlingstein, P., Gruber, N., Jones, S. D., Murray-Tortarolo, G., Ahlström, A., Doney, S. C., Graven, H., Heinze, 20 C., Huntingford, C., Levis, S., Levy, P. E., Lomas, M., Poulter, B., Viovy, N., Zaehle, S., Zeng, N., Arneth, A., Bonan, G., Bopp, L., Canadell, J. G., Chevallier, F., Ciais, P., Ellis, R., Gloor, M., Peylin, P., Piao, S., Le Quéré, C., Smith, B., Zhu, Z. and Myneni, R.: Trends and drivers of regional sources and sinks of carbon dioxide over the past two decades, Biogeosciences Discuss.,10(12), 20113–20177, doi:10.5194/bgd-10-20113-2013, 2015.
could be changed to
Sitch, S., Friedlingstein, P., Gruber, N., Jones, S. D., Murray-Tortarolo, G., Ahlström, A., Doney, S. C., Graven, H., Heinze, 20 C., Huntingford, C., Levis, S., Levy, P. E., Lomas, M., Poulter, B., Viovy, N., Zaehle, S., Zeng, N., Arneth, A., Bonan, G., Bopp, L., Canadell, J. G., Chevallier, F., Ciais, P., Ellis, R., Gloor, M., Peylin, P., Piao, S., Le Quéré, C., Smith, B., Zhu, Z. and Myneni, R.: Trends and drivers of regional sources and sinks of carbon dioxide over the past two decades, Biogeosciences, 12, 653–679, doi:10.5194/bg-12-653-2015, 2015.

I would also recommend to check carefully the list of the authors. It seems to me that "Ito Akihiko" should be changed to "Akihiko Ito" at the Page 1, line 5

**Thank you for the catching this error and noting a need for reference updates. We have made the suggested changes in the text.**

Reviewer #2

Review of the article "Role of CO2, climate and land use in regulating the seasonal amplitude increase of carbon fluxes in terrestrial ecosystems: a multimodel analysis" by Zhao et al.
This article presents interesting results in the scope of BG. It can be published after the authors haven taken care of the minor issues stated below. Therefore, I've asked for minor revisions only. However, I'm convinced that the paper could be improved substantially by a small effort, if the discussion would be extended in the two directions I try to describe in the following:
The study has two parts. In the first part it is evaluated, if nine global vegetation models (from the TRENDY project) can reproduce the seasonal cycle of land carbon fluxes as derived from atmospheric inversions (figure 1 to 3). In the second part of the article these models are used to investigate trends in the seasonal cycle amplitude of land carbon fluxes. This is done by separating the contributions to the trends for three major forcing factors: rising atmospheric CO2 concentration, trends in climate, and land use/cover change (figure 5 to 9).

**Thank you for the positive feedback and nice summary.**

In view of the large uncertainties in land carbon cycle model results, I appreciate this study very much. Combining model evaluation and factorial analysis will probably provide a better understanding of the seasonal cycle carbon fluxes (as also mentioned by the authors in the introduction, page 3 line 20-21). Unfortunately, this potential is not utilized. The findings about the performance of each model are (almost) not mentioned in the discussion of the second part. For example, the results shown in figure 5 could be discussed in view of the evaluation presented in figure 1. Some questions like the following could be posed and answered: are the models that successfully simulate the seasonal cycle of carbon uptake more similar in the CO2 fertilization factor than the models that fail to reproduce the seasonal cycle? Or is this true for the climate factor? Analogously, figure 6 could be compared with figure 2.

**Thanks for pointing this out. This aspect was indeed overlooked in the original version of this paper. In the revised version, we have incorporated some interesting similarity among the four models with similar mean seasonal cycle of global carbon flux inversions. Specifically, we added the following:**

**1) Global trend: added this paragraph on P9 before section 3.3.1:**
**"The four models (CLM4.5BGC, VEGAS, LPX-Bern and ORCHIDEE) that simulate a more realistic mean global $F_{TA}$ seasonal cycle (Figure 1) are also relatively close in global $F_{TA}$ seasonal amplitude, clustering around an increase of 14±3% during 1961-2012. Furthermore, they all suggest land use/cover change contribute**

**positively to global $F_{TA}$ seasonal amplitude increase. On the other hand, four of the remaining five models (OCN, LPJ, JULES, VISIT) show a much larger rate of increase (26±3%), but given the fact that these four models underestimate the mean amplitude by about 50%, the absolute increase in global $F_{TA}$ seasonal**

5      **amplitude is actually similar (about 5 PgC y$^{-1}$) between the two groups of models. ISAM is an exception, it both underestimates the mean global $F_{TA}$ seasonal amplitude and has the lowest rate of amplitude increase."**

         **2) Consistency in the climate factor for northern temperate region, change the original sentence on P9 in L5 to:**

10     **"In the Northern temperate (23.5-50N) region, climate change alone would decrease the $F_{TA}$ amplitude—this is consistent among the four models with realistic mean global and Northern temperate (Figure 2) $F_{TA}$ seasonal cycle simulation, but is not the case for JULES and LPJ (Figure 6), possibly related to mid-latitude drought (Buermann et al., 2007). "**

15     **3) At the end of section 3.3.3:**

    **"While most models indicate land use/cover change in Southern tropics (Amazon is probably the most notable region) decrease global $F_{TA}$ amplitude during 1961-2012, LPJ suggests it would cause a large increase in the amplitude instead, possibly related to its different behavior in simulating mean seasonal cycle of**

20     **carbon flux for that region (Figure 2d)."**

Finally, we want to understand by such studies, why the magnitude of the seasonal cycle in atmospheric CO2 increased. Additionally, we want to confirm that vegetation models respond reasonably to global warming and increasing atmospheric CO2. The

25     self-evident way to reach both aims is to evaluate models and then to analyse their results by taking into account this evaluation. The later step is incomplete in the current version of the text as the evaluation is not considered in the discussion of the forced simulation results.

30     **We have now incorporated evaluation of mean seasonal cycle in discussing the factorial attribution results as explained above, and this hopefully make the revised version more complete. We cannot confirm if all vegetation models respond reasonably to global warming and increasing CO2, as there are important differences in models' sensitivity to them (therefore not all models can respond**

35     **realistically to changes of CO2 and climate in all regions), especially at regional level, as also indicated from the results in this study. In the concluding remarks of this manuscript, we have outlined the future study that we would be very much interested to pursue, and encourage the community to work on. Such work will build on the important regional differences identified in this study and would**

40     **allow the models to be evaluated extensively and comprehensively, which would help to reach both of the suggested aims.**

Another important comment I would like to add concerns figure 10. This shows a moderate correlation between the change in net land carbon uptake and the increase in the amplitude of the seasonal carbon fluxes as simulated by the different vegetation models. The authors mention that this cross-model correlation may be used to constrain land carbon uptake (page 11 line 9). I think, this is the key motivation to investigate the seasonal amplitude of carbon fluxes and it should be also mentioned in the introduction. Furthermore, the authors claim for more research on observed CO2 fluxes and atmospheric transport on a regional scale to substantiate this finding (page 11, line 11-13). I agree, but the obvious next step is to further investigate the results of the model ensemble, how the correlation is simulated. A factorial analysis of the long-term carbon uptake should be performed and then compared with figure 5. Thereby, it could be specified, which factor contributes to what extent to the correlation. Probably this is beyond the scope of the article, but at least this opportunity should be mentioned. And, this kind of analysis is commonly denoted by the keyword "emergent constraint". It would be good to cite a reference, perhaps the paper of Cox et al 2013.

**The idea of using the keyword "emergent constraint" was indeed discussed among co-authors at an earlier stage of the paper, however we decided against it as we have not completely understood the mechanisms behind this correlation, and we already have lots of material in this paper at that point. The obvious next step as you mentioned is indeed to also perform a factorial analysis of the long-term carbon uptake, however since figure 10 does not actually represent the key of this paper, but rather a very interesting observation (similar observations were also made in Ito et al. 2016; Zhao and Zeng 2014) that has potential in future studies. We are very much interested to further explore this opportunity later, but for now, we decided to simply mention this as suggested after "in aggregated global values":**
**"A factorial analysis of the long-term carbon uptake could help to determine which factor contributes to what extent to this correlation."**

Minor scientific issues
- Why is the exceptional result of VEGAS concerning the CO2 factor mentioned in the abstract (page 1 line 32)? It is not a key finding.

**We do believe the result of VEGAS is important to be mentioned here since it is the only model that both simulates a realistic seasonal cycle of carbon flux and indicating CO2 is not the most important factor in the amplitude increase. This is possibly also associated with the weaker CO2 fertilization effect in VEGAS, whereas CO2 fertilization effect is strong in many other models. There are certainly many arguments on the strength of CO2 fertilization in real world, which is one of the most important issues in carbon cycle science but beyond the scope of this paper. Without including this key disagreement, the readers could be left only with the impression that a majority DGVMs agree on CO2 being the most important factor, which would be the opposite of what this paper tries to convey: the models disagree on the importance of the three factors, despite that they**

**generally agree on the overall trend, and we should validate them at regional scale in future studies.**

- "is a good indicator of terrestrial ecosystem dynamics" (page 2 line 11). This is a too general statement. It does not help in the line of argument. I would skip it.

**Deleted as suggested.**

- It would be nice to mention in section 2.1 that the TRENDY model simulations are offline simulations driven by climate data (and other input like atm. CO2 concentration) and that the models are not coupled to general circulation models. Of course this can be deduced from appendix A, but it should be also clearly stated in the main text as it is important for its perceivability (e.g. the differences in the results between the models can not be due to weather noise).

**Good suggestion. The sentence in section 2.1 now reads:**
**"A set of three offline experiments driven by either constant or varying climate data and other input such as atmospheric $CO_2$ and land use/cover forcing were designed in the TRENDY project to differentiate the role of $CO_2$, climate and land use (Table 2)."**

- The authors assume that the models simulate the effect of CO2, climate, and land use "linearly" (page 5 line 16-22). I think, "linear" is missleading in this context. It is more about synergy terms. For example, the trend in S2 minus S1 includes the climate effect and the synergy of CO2+climate. Furthermore, I'm not convinced that the synergy terms are all unimportant. Therefore, I propose to simply state that the climate effect and the synergy of CO2+climate together are called "climate" for simplicity in the rest of the manuscript. Without a discussion whether the synergy terms are negligible or not. And, of course, analogously for S3 minus S2.

**Good point. Even though we have tested that the synergy terms are very small in one model, we do not have the means to test that for all models. Following your suggestion, we have changed the relevant text as below:**
**"The effect of $CO_2$ on the relative amplitude change is represented by trend of S1 ($CO_2$ only) results, the S2 ($CO_2$+Climate) results show a trend that is the sum of $CO_2$ and climate effects, and the S3 ($CO_2$+Climate+Land Use/Cover) simulations include trends from time-varying $CO_2$, climate and land use/cover change (abbreviated as LandUse for text and figures). For simplicity, the effect of "climate" as used in this paper includes the synergy of $CO_2$ and climate, and similarly the effect of "land use/cover" also includes the synergy terms. Therefore, effect of $CO_2$, climate and land use/cover are then quantified as the trend for S1, trend of S2 minus S1 trend, and trend of S3 minus S2 trend, respectively. Note that the synergy terms are likely small in some of the current generation dynamic vegetation models, such as shown in previous sensitivity experiment results (Zeng et al., 2014)."**

- Please replace "Q10 value" (page 6 line 22) by "temperature dependence of heterotrophic respiration". Not everyone is familiar with this shortcut.

**Replaced as suggested.**

- Concerning the temporal trend in the seasonal amplitude in the late 90s (page 8 line 9): I can not deduce from figure 7 that half of the models exhibit a decrease, but it is obvious that the model ensemble shows an increase.

10 **We agree that the model ensemble obviously shows an increase, however that is largely contributed by the JULES model. For the rest of the models it is more likely a half-half split. Also "trend" is probably not the best word here to describe a change in a few years. Therefore, we have now stated the model names (LPJ, OCN, ORCHIDEE, VEGAS) where at least some sort of decrease in the late 90s is**
15 **found, even though the amplitude of this change maybe small and there is a rebound in some that is larger than the observation records indicate. We have also changed the word "trend" to "change". However we would like to avoid too detailed discussion here.**

- The models agree in the general seasonal amplitude increase, but they disagree in the contribution of the climate factor as well as the land use factor to the seasonal amplitude trend (see figure 5). I think, this disagreement even in sign should be mentioned in section 3.3. In the following subsections this important fact may be overseen due to all
25 the details stated there.

**Thanks for pointing this out. This message should indeed be put in a more prominent location. Therefore, the first sentence in section 3.3 is now changed to: "Models agree on increase of global $F_{TA}$ seasonal amplitude during 1961-2012, but**
30 **they disagree even in sign in the contribution of the different factors (Figure 5)."**

Typos etc.
page 1 line 25: replace "during 1961-2012 for its seasonal cycle and amplitude trend" by "for its seasonal cycle and amplitude trend during 1961-2012"
35 page 1 line 31: replace "is a stronger" by "is the strongest"
page 2 line 9: replace "of CO2 seasonal cycle" by "of atmospheric CO2 seasonal cycle"
page 2 line 25: replace "in understanding the contribution of various mechanisms" by "to disentangle effects of various mechanism"
page 3 line 1: replace "instead of dynamic vegetation models" by "instead of biases in
40 dynamic vegetation models"
page 4 line 14: replace "A direct comparison with fluxes from process-based models are monthly" by "Fluxes from process-based models can be directly compared with monthly"
page 4 line 29: replace "The seasonal amplitude of Mauna Loa Observatory or global
45 CO2 growth rate and fluxes from model simulations and inversions are computed" by "The seasonal amplitude at Mauna Loa Observatory, global CO2 growth rate, and fluxes from model simulations and inversions are processed"

page 5 line 2: replace "as seasonal amplitude" by "to define the seasonal amplitude"

page 5 line 28: replace "was defined in Eq. (1):" by "is defined as:"

page 11 line 33: replace "models' mechanical difference" by "the different parametrisations

of important processes in models"

page 12 line 21: replace "high latitude "greening" over high latitudes" by "high latitude "greening""

page 12 line 24: replace "to differ for different models" by "to differ between models"

page 27: replace "in the S2 experiment (changing $CO_2$ and climate and land use/cover) substracted by trends in S1 (changing $CO_2$ only)" by "in the S3 experiment (changing $CO_2$, climate, and land use/cover) substracted by trends in S2 (changing $CO_2$ and climate)"

Please proof articles. They are missing quite often.

**Many thanks for your careful checking, we have corrected all the above mentioned typos (in a few cases not exactly as suggested, in order to retain original meaning). We also did another round of proofing and made numerous minor changes in the text.**

[revised manuscript text omitted]

15    among the models. The effect of $CO_2$ on the relative amplitude change is represented by  trend of S1 ($CO_2$ only) results, the S2 ($CO_2$+Climate) results show a trend that is the sum of $CO_2$ and climate effects, and the S3 ($CO_2$+Climate+Land Use/Cover) simulations include trends from time-varying $CO_2$, climate and land use/cover change (abbreviated as LandUse for text and figures). For simplicity, the effect of "climate" as used in this paper includes the synergy of $CO_2$ and climate, and similarly the effect of "land use/cover" also includes the synergy terms.

20    Therefore, effect of $CO_2$, climate and land use/cover are then quantified as the trend for S1, trend of S2 minus S1 trend, and trend of S3 minus S2 trend, respectively. Note that the synergy terms are likely small in some of the current generation dynamic vegetation models, such as those shown in previous sensitivity experiment results (Zeng et al., 2014).

**2.5 Spatial attribution**

25    Spatial attribution of global $F_{TA}$ amplitude change can be difficult due to the phase difference at various latitudes. For example, the two amplitude peaks at Northern and Southern subtropics caused by monsoon movements are largely out of phase, and the net contribution to global $F_{TA}$ amplitude increase after their cancelation is small (Zeng et al., 2014). To quantify latitudinal and spatial contributions for each model, a unique quantity—$F_{k\_A}^{i}$, difference between the maximum month ($i\_max$) and the minimum month ($i\_min$) of model $i$'s global $F_{TA}$, based on model $i$'s

30    2001-2010 mean seasonal cycle  is defined in Eq. (1):

$$F_{k\_A}^{i} = F_{k\_A(i\_max)}^{i} - F_{k\_A(i\_min)}^{i} \,,$$
$$(1)$$

The subscript $k$ denotes index  of each latitudinal band or spatial grid, and $A$ is index of year, ranging from 1961

35    to 2012. $F_{k\_A}^{i}$ could be quite different for each model: for VEGAS, $F_{k\_A}^{i}$ is $F_{TA}$ in November ($i\_max$ = 11) minus $F_{TA}$

in July ($i\_min = 7$) in year $A$, and for LPJ, $F_{k\_A}^{i}$ is $F_{TA}$ in March ($i\_max = 3$) minus $F_{TA}$ in June ($i\_min = 6$) in year $A$. The indexes $i\_max$ and $i\_min$ are fixed for each model, as summarized in Table 3. For all three experiments, $F_{k\_A}^{i}$ is computed each year in 1961-2012 and at every latitude band or spatial grid ($k$), and then the trends of $F_{k\_A}^{i}$ are calculated. The spatial aggregation of the resulted latitudinal-depended trends would then approximately equal to trend of global $F_{TA}$ maximum-minus-minimum seasonal amplitude.

**3 Results**

**3.1 Mean seasonal cycle of $F_{TA}$**

Four of the nine models (CLM4.5BGC, LPX-Bern, ORCHIDEE and VEGAS) simulate a mean global $F_{TA}$ seasonal cycle of similar amplitude and phase compared with the Jena99 and CarbonTracker inversions (Figure 1, Table 3). The other five models have much smaller seasonal amplitude than inversions, and the shape of the seasonal cycle is also notably different. As a result, models' ensemble global $F_{TA}$ has seasonal amplitude of 26.1 PgC y$^{-1}$ during 2001-2010, about 40% smaller than the inversions (Figure 4 inset, Table 3). The model ensemble annual mean $F_{TA}$ (residual land sink plus land use emission) is −1.1 PgC y$^{-1}$ for 2001-2010, 30% smaller than the inversions (Table 3). In some models (ISAM, JULES, and LPJ for the Northern Temperature region in Figure 2) $F_{TA}$ rebounds back quickly, resulting in a late summer $F_{TA}$ maximum. The mid-summer rebound is unlikely a model response to pronounced seasonal drought after 2000, as it is persistent in the mean seasonal cycle over every decade since 1961. A probable cause is the strong exponential response of soil respiration to temperature increase, which may lead to heterotopic respiration higher than NPP in summer. For example, HadGEM2-ES and HadCM3LC that employ a forerunner of JULES3.2 used in this study, are found to have a comparatively better simulation of the seasonal cycle (Collins et al., 2011), due to a combination of a more sensitive temperature rate modifier combined with a larger seasonal soil temperature that are used in the later version of JULES. the HadCM3LC that employs TRIFFID, an earlier version of JULES3.2 used in this study, is found to have a large mid-summer peak carbon release over temperate North America (Cadule et al., 2012). Alexandrov (2014) shows that both the amplitude underestimation and phase shift of $F_{TA}$ seasonal cycle can be improved by increasing water use efficiency, decreasing Q10 valuetemperature dependence of heterotrophic respiration, and increasing the share of quickly decaying litterfall. Another probable factor is the simulation of plant phenology. With the help of remote sensing data, better phenology in model simulation has been shown to improve seasonal cycle simulation of carbon flux (Forkel et al., 2014). Additionally, the effect of carbon release from crop harvest is considered. If harvested carbon is the main cause for the mid-summer rebound in some models, the rebound should be much less pronounced for the S2 (constant 1860 land use/cover) experiment, given that cropland area in 1860 is less than half of the 2000 level. However, based on the comparison between the S2 and S3 experiments over global and northern temperate (major crop belts) $F_{TA}$ seasonal cycle (Figure S1 and S2), the impact of harvested carbon flux is unlikely to explain the mid-summer rebound. This is probably due to modeling efforts to prevent the sudden release of harvested carbon. Instead, carbon release of harvested products and/or their residuals is usually either spread over 12 months (i.e., LPJ, LPX-Bern, OCN, ORCHIDEE) or enters soil litter carbon pool (i.e., ISAM) for subsequent decomposition over time.

[revised manuscript text omitted]

The four models (CLM4.5BGC, VEGAS, LPX-Bern and ORCHIDEE) that simulate a more realistic mean global $F_{TA}$ seasonal cycle (Figure 1) are also relatively close in global $F_{TA}$ seasonal amplitude, clustering around an increase of 14±3% during 1961-2012. Furthermore, they all suggest land use/cover change contribute positively to global $F_{TA}$ seasonal amplitude increase. On the other hand, four of the remaining five models (OCN, LPJ, JULES, VISIT) show a much larger rate of increase (26±3%), but given that these four models underestimate the mean amplitude by about 50%, the absolute increase in global $F_{TA}$ seasonal amplitude is actually similar (about 5 PgC $y^{-1}$) between the two groups of models. ISAM is an exception: it both underestimates the mean  global $F_{TA}$ seasonal amplitude and has the lowest rate of amplitude increase.

**3.3.1 The rising CO2 factor**

Seven of the nine models suggest that $CO_2$ fertilization effect is most responsible for the increase in the amplitude of global $F_{TA}$, while VEGAS attribute it approximately equal among the three factors (Figure 5). The $CO_2$ fertilization effect alone seems to cause most of the amplitude increase in a majority of models, with notable contribution from climate change and land use/cover change in CLM4.5BGC and VEGAS (Figure 7). The effect of rising $CO_2$ appears to be slightly negative for JULES, possibly reflecting an offsetting of the strong seasonal soil respiration response

[revised manuscript text omitted]